# Effects of domestication on the gut microbiota parallel those of human industrialization

Aspen T Reese[1,2†]*, Katia S Chadaideh[1], Caroline E Diggins[1], Laura D Schell[1], Mark Beckel[3], Peggy Callahan[3], Roberta Ryan[3], Melissa Emery Thompson[4], Rachel N Carmody[1]*

[1]Department of Human Evolutionary Biology, Harvard University, Cambridge, MA, United States; [2]Society of Fellows, Harvard University, Cambridge, MA, United States; [3]Wildlife Science Center, Stacy, MN, United States; [4]Department of Anthropology, University of New Mexico, Albuquerque, NM, United States

**Abstract** Domesticated animals experienced profound changes in diet, environment, and social interactions that likely shaped their gut microbiota and were potentially analogous to ecological changes experienced by humans during industrialization. Comparing the gut microbiota of wild and domesticated mammals plus chimpanzees and humans, we found a strong signal of domestication in overall gut microbial community composition and similar changes in composition with domestication and industrialization. Reciprocal diet switches within mouse and canid dyads demonstrated the critical role of diet in shaping the domesticated gut microbiota. Notably, we succeeded in recovering wild-like microbiota in domesticated mice through experimental colonization. Although fundamentally different processes, we conclude that domestication and industrialization have impacted the gut microbiota in related ways, likely through shared ecological change. Our findings highlight the utility, and limitations, of domesticated animal models for human research and the importance of studying wild animals and non-industrialized humans for interrogating signals of host–microbial coevolution.

**\*For correspondence:**
areese@ucsd.edu (ATR);
carmody@fas.harvard.edu (RNC)

**Present address:** †Ecology Behavior & Evolution Section, University of California San Diego, La Jolla, United States

**Competing interests:** The authors declare that no competing interests exist.

## Introduction

Industrialized, agrarian, and foraging human populations differ along numerous ecological dimensions, including diet, physical activity, the size and nature of social networks, pathogen exposure, types and intensities of medical intervention, and reproductive patterns. Such changes have resulted in large shifts in the gut microbiota in industrialized populations relative to non-industrialized populations or closely related primates (*De Filippo et al., 2010*; *Moeller et al., 2014*; *Moeller, 2017*; *Smits et al., 2017*), including reductions in alpha-diversity and changes in composition that have been implicated in the rise of various metabolic and immunological diseases (*Ley et al., 2006*; *Cox et al., 2014*; *Kamada et al., 2013*). Many aspects of ecology that now differ between industrialized and non-industrialized human populations were similarly altered during the process of animal domestication (*Zeder, 2012*). For example, domestic animals often consume less diverse, more easily digestible diets than their wild relatives, expend less energy to achieve adequate (or excess) caloric intake, live in comparatively static and high-density groups, and can be subject to modern medical interventions including antibiotic treatment (*McClure, 2013*). Although industrialization and domestication are fundamentally different processes, the ecological parallels between human industrialization and animal domestication suggest that the gut microbiota of diverse domesticated animals may differ in consistent ways from those of their wild progenitors, and further, that their

**eLife digest** Living inside our gastrointestinal tracts is a large and diverse community of bacteria called the gut microbiota that plays an active role in basic body processes like metabolism and immunity. Much of our current understanding of the gut microbiota has come from laboratory animals like mice, which have very different gut bacteria to mice living in the wild. However, it was unclear whether this difference in microbes was due to domestication, and if it could also be seen in other domesticated-wild pairs, like pigs and wild boars or dogs and wolves.

A few existing studies have compared the gut bacteria of two species in a domesticated-wild pair. But, studies of isolated pairs cannot distinguish which factors are responsible for altering the microbiota of domesticated animals. To overcome this barrier, Reese et al. sequenced microbial DNA taken from fecal samples of 18 species of wild and related domesticated mammals.

The results showed that while domesticated animals have different sets of bacteria in their guts, leaving the wild has changed the gut microbiota of these diverse animals in similar ways. To explore what causes these shared patterns, Reese et al. swapped the diets of two domesticated-wild pairs: laboratory and wild mice, and dogs and wolves. They found this change in diet shifted the gut bacteria of the domesticated species to be more similar to that of their wild counterparts, and vice versa. This suggests that altered eating habits helped drive the changes domestication has had on the gut microbiota.

To find out whether these differences also occur in humans, Reese et al. compared the gut microbes of chimpanzees with the microbiota of people living in different environments. The gut microbial communities of individuals from industrialized populations had more in common with those of domesticated animals than did the microbes found in chimpanzees or humans from non-industrialized populations. This suggests that industrialization and domestication have had similar effects on the gut microbiota, likely due to similar kinds of environmental change.

Domesticated animals are critical for the economy and health, and understanding the central role gut microbes play in their biology could help improve their well-being. Given the parallels between domestication and industrialization, knowledge gained from animal pairs could also shed light on the human gut microbiota. In the future, these insights could help identify new ways to alter the gut microbiota to improve animal or human health.

differences may resemble those observed between industrialized and non-industrialized human populations.

Many of the altered ecological features experienced by industrialized humans and domesticated animals have been independently observed to impact the gut microbiota, including diet (*David et al., 2014*; *Carmody et al., 2015*), physical activity (*Allen et al., 2018*; *Lamoureux et al., 2017*), the size and nature of social networks (*Dill-McFarland et al., 2019*; *Antwis et al., 2018*), antibiotic use (*Bokulich et al., 2016*; *Cho et al., 2012*), and changes in birthing and lactation practices (*Bokulich et al., 2016*; *Li et al., 2018*). The effects of these features on gut microbiota composition are often found to match or exceed the effects of genetic variation (*Carmody et al., 2015*; *Rothschild et al., 2018*), which is also routinely modified by domestication. As such, ecological shifts under domestication might be expected to lead to gut microbial differentiation between domesticated animals and their wild counterparts. To this end, wild mice have been shown to differ from laboratory mice in gut microbial composition (*Kreisinger et al., 2014*; *Rosshart et al., 2017*). Similarly, a comparison of domesticated horses and wild Przewalski's horses in adjacent Mongolian grasslands found that the wild animals harbored compositionally distinct, and overall more diverse, gut microbial communities (*Metcalf et al., 2017*). However, to date, no general survey has been conducted to characterize the global effects of domestication on the gut microbiota.

Apart from the pressures of ecological change that domestic animals experience in human environments, animal domestication has also entailed strong artificial selection for phenotypes desirable to humans, such as rapid growth and docility in agricultural animals, reliable reproduction and stress resistance in laboratory animals, and unique physical and/or behavioral attributes in companion animals. Although targeted phenotypes differ based on the species under domestication, all domesticated mammals share the legacy of having been intentionally or indirectly selected for

tameness (*Wilkins et al., 2014*). This selection has been argued to have resulted in convergent morphological and physiological features across domesticated mammals that are collectively referred to as 'domestication syndrome' – including, for instance, reductions in brain size and tooth size, depigmentation, altered production of hormones and neurotransmitters, and retention of juvenile behaviors into adulthood – with the pleiotropic nature of these effects thought to be mediated by changes in neural crest cells (*Wilkins et al., 2014*). Therefore, to the extent that gut microbiota is dependent on host biology, we might additionally expect domestication to have shaped the gut microbiome in similar, potentially convergent, ways across diverse mammalian lineages. Such microbiota-structuring contributions ascribable to evolutionary rather than ecological forces have the potential to be much greater in and/or unique to domesticated animals relative to industrialized human populations since the process of domestication has been advancing for much longer than industrialization.

Here, we assess the effects of domestication on the mammalian gut microbiota, perform controlled dietary experiments that attempt to distinguish between the relative roles of ecology and genetics in driving these patterns, and compare the effects of domestication to those of human industrialization. While we focus primarily on the impacts of domestication on the mammalian gut microbiota, we include analyses of industrialized and non-industrialized human populations because much is known about the effects of industrialization on the gut microbiota and as such it can serve as a benchmark ecological process for domestication. In addition, to explore the extent to which deeper evolutionary history affects these patterns, we also compare humans to chimpanzees (*Pan troglodytes*), one of our two closest living relatives and arguably the better referential model for the last common ancestor between *Pan* and *Homo* (*Muller et al., 2017*). Early *Homo sapiens* is thought to have undergone a form of self-domestication as a result of selection against aggression (*Wrangham, 2018*; *Theofanopoulou et al., 2017*), suggesting that there could likewise be parallels between the gut microbial signatures of animal domestication and *Pan–Homo* speciation.

We predict that (i) gut microbial communities will differ between domesticated animals and their wild counterparts, (ii) gut microbial communities of diverse domesticated animals may exhibit convergent characteristics in a microbial counterpart to the physiological domestication syndrome (*Wilkins et al., 2014*), and (iii) gut microbial changes observed with domestication may parallel contrasts observed between chimpanzees and humans. In addition, to the extent that domestication effects are driven by ecology rather than host phylogenetic distance, we should expect (iv) experimental manipulation of ecology to overcome differences in the gut microbiota between closely related hosts, and (v) the gut microbiota of domesticated animals will share more features with industrialized human populations than with non-industrialized human populations.

Identifying the factors shaping the gut microbiota of domesticated animals will provide insights into the ecology of host-associated microbial communities and their impact on health. Domesticated animals serve as reservoirs for zoonotic diseases (*Morand et al., 2014*; *Wolfe et al., 2007*; *Cleaveland et al., 2001*; *Han et al., 2016*) and carriers of antibiotic-resistant bacteria (*Sayah et al., 2005*; *EFFORT Group et al., 2018*). Furthermore, the ecological impacts of domestication on the gut microbiota could conceivably contribute to the unique health problems experienced by captive (*Hosey et al., 2009*) and domesticated animals (*Timoney et al., 1988*). Differences between domesticated and wild animal microbiota may also manifest in poor translatability between laboratory studies and the real world (*Leung et al., 2018*; *Beura et al., 2016*). Finally, the convergent nature of many ecological shifts experienced by domesticated animals and industrialized human populations suggests that domesticated animals may provide a uniquely useful model for studying the microbially mediated health impacts of rapid environmental change that results in mismatch between host, microbiota, and/or environment, a situation thought to apply to humans in industrialized settings (*Sonnenburg and Sonnenburg, 2019*). Understanding what shapes the domesticated microbiota may therefore identify routes to improve experimental models, animal condition, and human health.

## Results

### Cross-species comparison of gut microbial composition

First, we characterized the fecal microbiota of wild and domesticated populations of nine pairs of artiodactyl, carnivore, lagomorph, and rodent species (*Figure 1A*) using 16S rRNA gene amplicon sequencing and quantitative PCR (qPCR). Despite observing no single convergent 'domesticated microbiota' profile, our analysis detected a global signal of domestication status on gut microbiota composition. Across the combined dataset, the factor that explained the largest proportion of variation was the host dyad (e.g., pig/boar; p<0.001, $R^2$ = 0.39, F = 17.086, permutational multivariate analysis of variance [PERMANOVA]; *Figure 1B*). However, correcting for host dyad, domestication status also contributed significantly to variation in microbial communities (p<0.001, $R^2$ = 0.15, F = 6.081), and these results were robust to the distance metric analyzed (*Supplementary file 1*). Furthermore, analyses of individual dyads found a significant effect of domestication status for all groups except canids (p<0.05, $R^2$ = 0.18–0.41, PERMANOVAs; *Supplementary file 1*). Diet and digestive physiology were also primary determinants of the gut microbiota (diet: p<0.001, $R^2$ = 0.12, F = 21.216; physiology: p<0.001, $R^2$ = 0.14, F = 23.938; *Figure 1—figure supplement 1*), as seen in other surveys of mammals (*Muegge et al., 2011*), with effect sizes comparable to that of domestication status. Consistent with the idea that higher ecological homogeneity in domesticates may beget greater gut microbial homogeneity, we found that there was greater between-conspecific variability in wild gut communities than in domesticated gut communities (p=0.002, F = 8.838; permutation test for F).

To determine whether there was a consistent change in microbial composition with domestication, we calculated the difference between an individual's ordination coordinates and the average ordination coordinates of its host dyad along the first nonmetric multidimensional scaling (NMDS) axis. Quantifying this ordination shift allowed us to consider overall changes in composition while

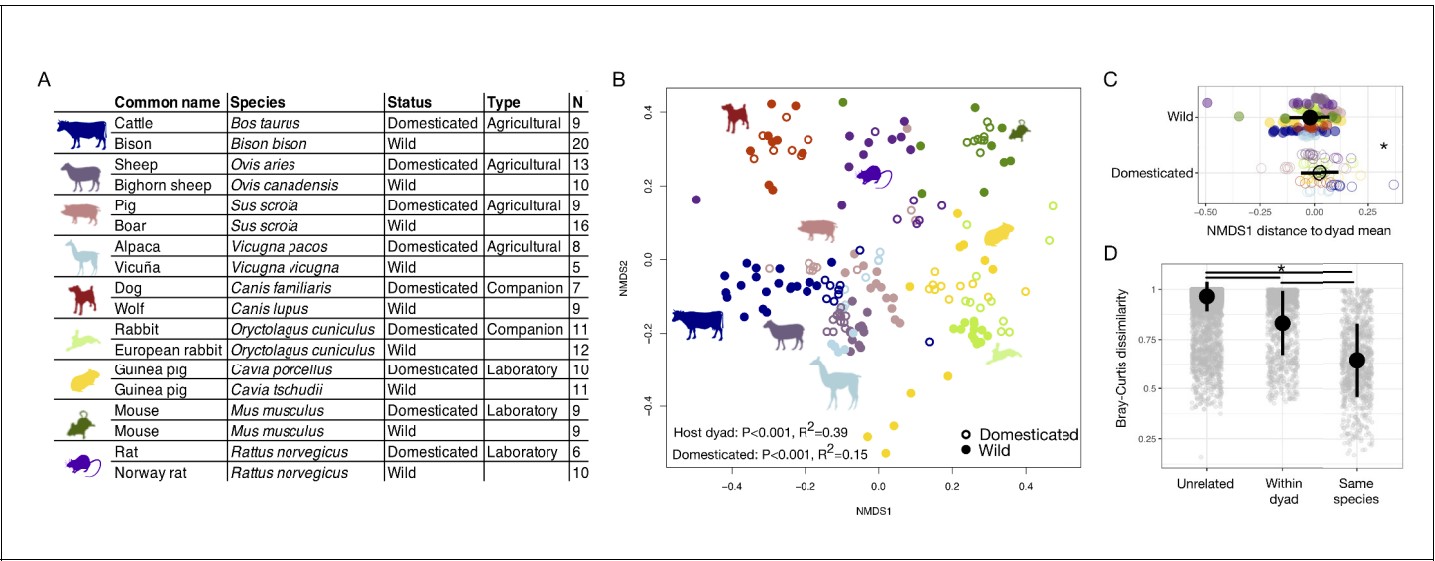

**Figure 1.** The mammalian gut microbiota carries a global signature of domestication. (**A**) Sampling scheme for cross-species study. (**B**) Nonmetric multidimensional scaling (NMDS) ordination of Bray–Curtis dissimilarities illustrates a significant signal (p<0.001, $R^2$ = 0.15, F = 6.081, permutational MANOVA) of domestication (closed versus open circles; N = 82 domesticated and 99 wild) and clustering by host dyad (color; N = 5–20 individuals per species). (**C**) Distance to dyad (color) mean along Bray–Curtis ordination NMDS axis 1 differs by domestication status (p=0.006, Mann–Whitney U test). (**D**) Bray–Curtis dissimilarity between individuals is lowest among conspecifics, but wild–domesticated pairs also have lower dissimilarity than unrelated pairs (p<0.001, bootstrapped Mann-Whitney U tests). * indicates p<0.05. Large circles are means; bars show standard deviations. The online version of this article includes the following figure supplement(s) for figure 1:

**Figure supplement 1.** Alternative factors associated with gut microbiota variation among wild and domesticated mammals.

**Figure supplement 2.** Microbial parameter comparisons between wild and domesticated mammals and between chimpanzees and humans.

**Figure supplement 3.** Relatedness was correlated with ordination shifts but not dissimilarity within a dyad.

**Figure supplement 4.** Domestication explained few global gut microbial community characteristics in our cross-species dataset.

correcting for host dyad and retaining information on the directionality of changes. We found that domesticated individuals were typically further right relative to the average of their host dyad (p=0.006, Mann–Whitney U test; *Figure 1C*) and that this difference was significant regardless of the distance metric analyzed (*Supplementary file 1*). Most domesticated species displayed similar trends in these ordination shifts (*Figure 1—figure supplement 2*) with laboratory and companion animals showing significant differences when analyzed collectively (p<0.05, Mann–Whitney U tests; *Figure 1—figure supplement 2*).

Free-ranging wild animal populations representing the progenitor species were not sampled for all pairs, potentially limiting the scope of our analysis. To assess whether the patterns described held for more stringent groupings, we analyzed the subset of wild–domesticated dyads for which the wild member was from a free-ranging population (i.e., 'truly wild') as well as the subset for which the wild member was the known progenitor (i.e., 'perfect pair') (*Supplementary file 2*). In both cases, we still found that domestication status explained a meaningful portion of variation in gut microbial community composition, regardless of the distance metric used (all p<0.001, all $R^2$ > 0.11, PERMANOVA; *Supplementary file 1*).

Domestication of mammalian species occurred at different times, so the evolutionary relationships between members of a host dyad are not all equal, even in cases where we sampled the known progenitors. Supporting an underlying influence of host evolutionary history on the gut microbiota, we found that host species that were more closely related (i.e., had a shorter time since divergence) had more similar microbial community compositions (p<0.001, r = 0.157, Mantel test). Similarly, the magnitude of the ordination shifts along NMDS axis 1 were smaller for animals from host dyads that were more closely related (p=0.012, rho = 0.19, Spearman correlation; *Figure 1—figure supplement 3*). Nevertheless, supporting the idea that ecology plays a dominant role in shaping the gut microbiota, average dissimilarity between members of a dyad was not lower in species pairs with more recent dates of domestication (p=0.854) or more recent divergence times (p=0.380; *Figure 1—figure supplement 3*). Moreover, differences in the ordination shifts along NMDS axis 1 associated with domestication remained significant even when correcting for host dyad and divergence time (p<0.001, likelihood test linear mixed effects models). Overall, dissimilarity between conspecifics was lowest, but dissimilarity between wild–domesticated dyads was significantly lower than for unrelated pairs (p<0.001, bootstrapped Kruskal–Wallis tests; *Figure 1D*).

We also tested for differences in specific features of the gut microbiota between domesticated and wild mammals. Domestication status did not affect microbial density quantified as copies of the 16S rRNA gene per gram of feces (p=0.089, Mann–Whitney U test), Shannon index (p=0.2017), or operational taxonomic unit (OTU) richness (p=0.3506; *Figure 1—figure supplement 4*), indicating that the domestication signal overall was not primarily driven by microbial species loss. Consistent with experiencing heightened environmental exposure, wild animals generally harbored potential pathogen communities that were more diverse (p=0.001, Mann–Whitney U test) and marginally more abundant (p=0.092; *Figure 1—figure supplement 4*). Among laboratory animals specifically, potential pathogen abundance (p<0.001) and pathogen richness (p<0.001) were substantially lower than among wild relatives, while total microbial density was higher (p=0.006; *Figure 1—figure supplement 2*). Companion animals did not differ significantly by domestication status for microbial density, diversity, or pathogen metrics. By contrast, agricultural animals had higher Shannon index and richness values (p≤0.001, Mann–Whitney U tests) as well as marginally higher pathogen abundances (p=0.067; *Figure 1—figure supplement 2*) compared with their wild counterparts.

## Diet versus host taxon effects on domesticated gut microbial composition in mice

Domestication has had profound effects on both ecology and host genetics. To begin to tease apart the relative roles of ecological change and genetic change in shaping the gut microbiota in domesticates, we performed a series of reciprocal diet experiments that tested the extent to which gut microbial signatures of wild–domesticated dyads could be recapitulated and reversed solely by the administration of ecologically relevant diets. We first conducted a fully factorial experiment in which wild-caught and laboratory mice (*Mus musculus*) were maintained for 28 days on wild or domesticate diets (*Figure 2A*, *Supplementary file 3*). Overall, we found that host taxon explained the largest amount of variation in composition (p<0.001, $R^2$ = 0.173, F = 64.255, PERMANOVA), but that diet (p<0.001, $R^2$ = 0.042, F = 15.427) and a host taxon by diet interaction term (p<0.001, $R^2$ = 0.020,

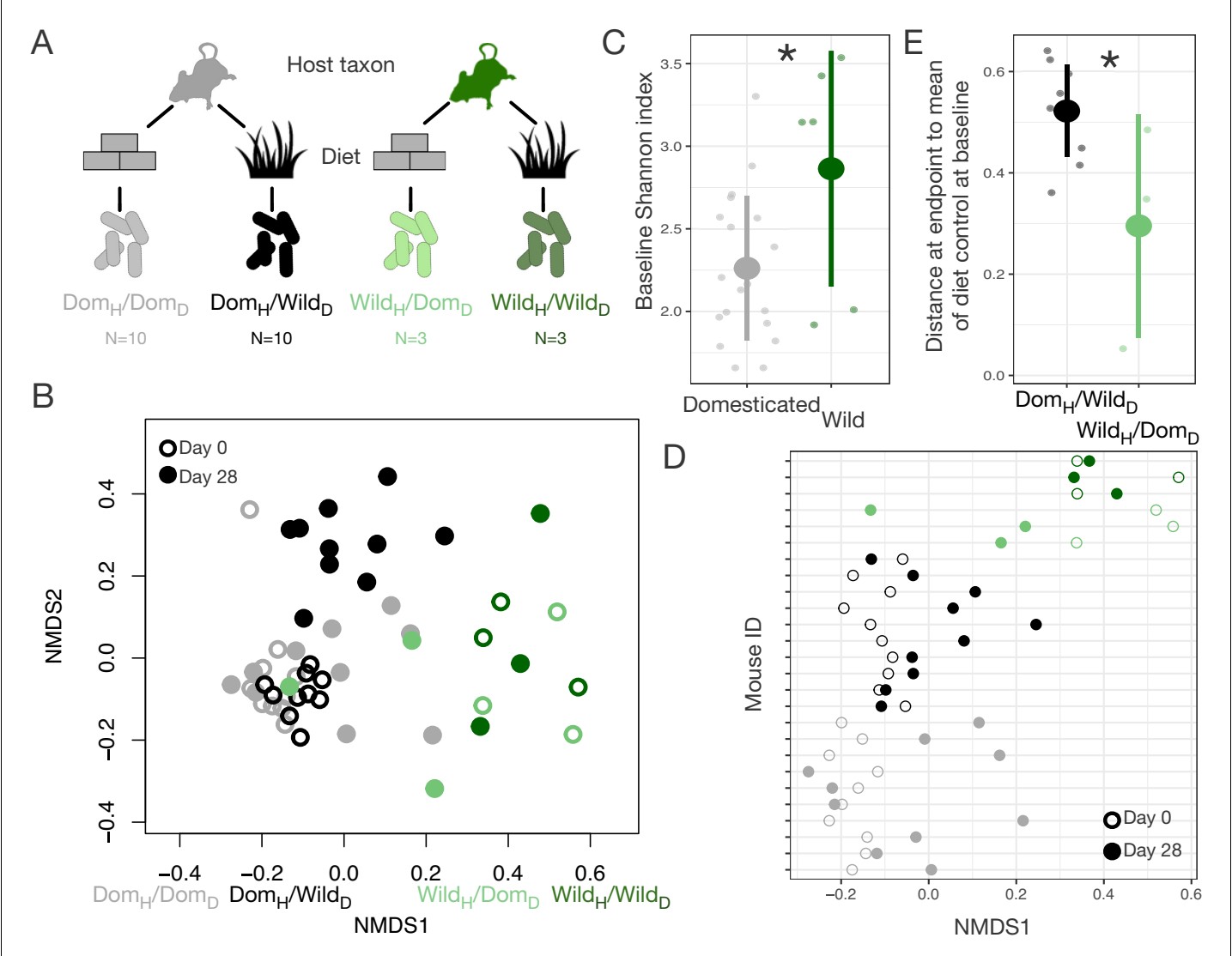

**Figure 2.** Gut microbial differences between wild and domesticated mice can be partially overcome by diet swap. (**A**) Design scheme for fully factorial host taxon by diet mouse experiment (N = 10 laboratory mice or three wild mice per diet group). (**B**) Nonmetric multidimensional scaling (NMDS) ordination of Bray–Curtis dissimilarities showing changes for mice from day 0 (open circle) to day 28 (filled circle) by experimental groups (color). Composition varied by host taxon (p<0.001, $R^2$ = 0.173, F = 64.255, permutational MANOVA), diet (p<0.001, $R^2$ = 0.042, F = 15.427), and a host taxon by diet interaction (p<0.001, $R^2$ = 0.020, F = 7.557). (**C**) Shannon index differed between host taxa on day 0 (p=0.011, Mann–Whitney U test). (**D**) Animals on reciprocal diets (Dom$_H$/Wild$_D$ and Wild$_H$/Dom$_D$) but not control diets tended to move in opposite directions along Bray–Curtis ordination NMDS axis 1 from day 0 to day 28 (p=0.048 and p=0.25, respectively, one-sample Wilcoxon test). (**E**) At the end of the experiment, distance to the mean of the diet control at baseline (Dom$_H$/Dom$_D$ and Wild$_H$/Wild$_D$) was lower for wild mice than for laboratory mice (p=0.048, Mann–Whitney U test). * indicates p<0.05, Mann–Whitney U test. Large circles are means; bars show standard deviations.

The online version of this article includes the following figure supplement(s) for figure 2:

**Figure supplement 1.** Microbiota composition at all time points during the mouse diet swap experiment.

**Figure supplement 2.** Select microbial and host metabolic parameters under mouse diet swap.

F = 7.557) were also significant (*Figure 2B*, *Figure 2—figure supplement 1*). Ordination shifts describing changes in the gut microbial community over the course of the experiment depended on the experimental group (axis 1: p<0.001, linear mixed effects model likelihood test; *Figure 2D*). Confirming prior reports that diet plays a dominant role in shaping the murine gut microbiota (*Carmody et al., 2015*), the gut microbiota of wild mice fed a domesticate diet (Wild$_H$/Dom$_D$) moved toward the average microbial community of domesticated mice fed a domesticate diet

(Dom$_H$/Dom$_D$), the microbiota of domesticated mice fed a wild diet (Dom$_H$/Wild$_D$) moved in the opposite direction, and those of control wild or domesticated mice consuming their habitual diets (Wild$_H$/Wild$_D$ and Dom$_H$/Dom$_D$) did not shift (*Figure 2B*). Over the course of the experiment, alpha-diversity as measured by Shannon index also changed significantly across treatment groups (p=0.025, Kruskal–Wallis test; *Figure 2—figure supplement 2*), with Dom$_H$/Wild$_D$ mice becoming significantly more diverse (p=0.004, one-sample Wilcoxon test) despite lower baseline levels of alpha-diversity in domesticated versus wild mice (p=0.011, Mann–Whitney U test; *Figure 2C*).

Neither host taxon nor diet was associated with differences in gut microbial density over the experiment (p=0.272, Kruskal–Wallis test; *Figure 2—figure supplement 2*), but it is notable that the total amount of feces produced, and thus likely the total number of bacteria, was lower in both host taxon groups when consuming the wild diet (p<0.001, Kruskal–Wallis test; *Figure 2—figure supplement 2*). Despite similar trends in fecal production between wild and domesticated mice in response to diet treatment, wild and domesticated mice differed markedly in their ability to harvest energy from experimental diets (p<0.001, Kruskal–Wallis test; *Figure 2—figure supplement 2*), as indexed by bomb calorimetry of feces. While wild mice were equally efficient digesters of the wild and domesticated diets, laboratory mice captured 15% fewer calories when consuming the wild versus domesticated diet.

Interestingly, asymmetries were also observed between wild and domesticated mice in their gut microbial responses to reciprocal diets. Whereas the microbial communities of Wild$_H$/Dom$_D$ mice grew to resemble those of untreated Dom$_H$/Dom$_D$ mice, the microbial communities of Dom$_H$/Wild$_D$ mice remained distinct from untreated Wild$_H$/Wild$_D$ mice throughout the experiment (p=0.042, Mann–Whitney U test; *Figure 2B, E*). It is possible that the asymmetry in energy harvest between wild and domesticated mice was rooted in differential microbial responses to reciprocal diets and the inability of Dom$_H$/Wild$_D$ mice to harbor a wild-type microbiota.

## Loss of wild gut microbiota in domesticated mice

Based on the lower alpha-diversity in domesticated versus wild mice (*Figure 2C*), we hypothesized that the asymmetries between domesticated and wild mouse responses to altered diets were due to past extinction of relevant strains from laboratory microbial communities and no dispersal source of replacement strains (*Sonnenburg et al., 2016*). Therefore, we next tested whether experimental dispersal from a wild microbial community in conjunction with feeding a wild diet could support a fully wild microbial community in laboratory mice (*Figure 3A*). A single colonization treatment with a wild mouse cecal community (via gavage) led to significant shifts in the gut microbial community (*Figure 3B*, *Figure 3—figure supplement 1*), resulting in closer resemblance to the wild donor (p<0.001, Mann–Whitney U test; *Figure 3C*). Shifts in NMDS axis 1 varied across experimental treatment groups (p<0.001, linear mixed effects model likelihood test; *Figure 3D*). While laboratory mice fed a wild diet but given a control gavage (PBS) also moved toward the donor along NMDS axis 1 (p=0.002, one-sample Wilcoxon test; *Figure 3D*), reflecting the influence of diet, we observed a substantially greater shift following experimental colonization (p<0.001, Kruskal–Wallis test). Surprisingly, among colonized mice, movement of the microbial community toward the wild donor profile was profound even without reinforcement from the wild diet (p=0.182, Mann–Whitney U test).

Although all mice exhibited an increase in microbial density over the course of the experiment (p<0.01, one-sample Wilcoxon tests), colonization with a wild community did not lead to higher microbial density overall (p=0.449, Kruskal–Wallis test; *Figure 3—figure supplement 1*) nor to an increase in alpha-diversity relative to baseline (p=0.258, one-sample Wilcoxon test). As in the original reciprocal diet experiment, wild diet treatment led to lesser fecal production (p<0.001, Kruskal–Wallis test; *Figure 3—figure supplement 1*). No differences in fecal production were observed between mice colonized with a wild community and PBS-treated controls (p=0.79; *Figure 3—figure supplement 1*), suggesting that lower fecal output on the wild diet was not a direct consequence of harboring a wild microbiota. Together, these results suggest that differences observed with experimental colonization reflected shifts in gut microbial community structure rather than simple augmentation of microbial load.

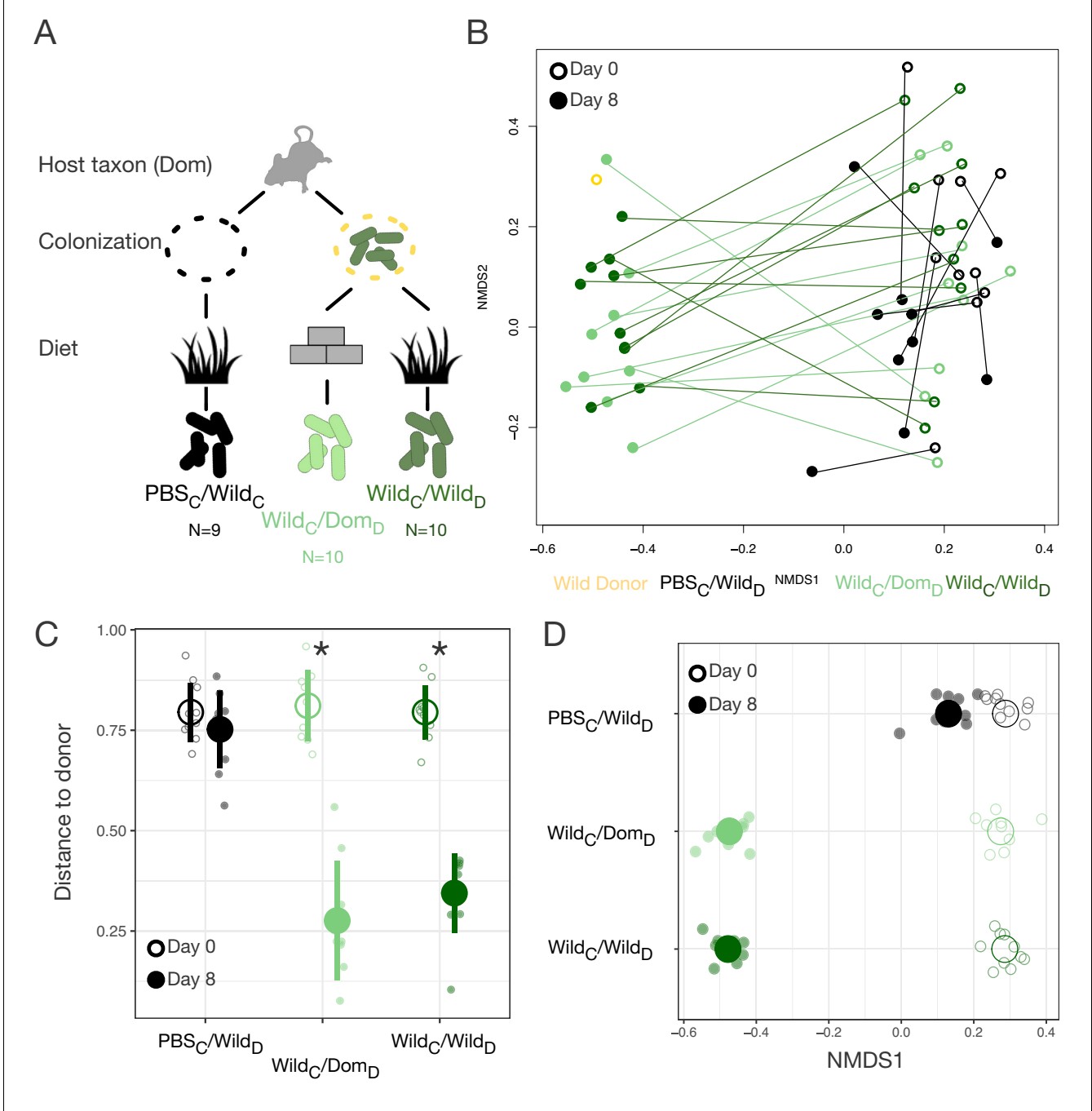

**Figure 3.** Laboratory mice can be re-wilded through colonization with a wild gut microbial community. (**A**) Design scheme for colonization/diet mouse experiment (N = 9–10 mice per treatment group). (**B**) Nonmetric multidimensional scaling (NMDS) ordination of Bray–Curtis dissimilarities showing changes for mice from day 0 (open circles) to day 8 (filled circles) by experimental groups (color). (**C, D**) At the end of the experiment (filled circle), distance to the wild community donor decreased most in animals colonized with wild communities (p=0.004 $Wild_C/Dom_D$ and p=0.002 $Wild_C/Wild_D$, Mann–Whitney U test; **C**), but all experimental groups exhibited change along Bray–Curtis ordination NMDS axis 1 (p=0.002 $PBS_C/Wild_D$, p=0.004 $Wild_C/Dom_D$, and p=0.002 $Wild_C/Wild_D$, one-sample Wilcoxon tests; **D**) during the course of the experiment. * in (**C**) indicates p<0.05, Mann–Whitney U test comparing day 0 to day 8 for each experimental group. Large circles are means; bars in (**C**) show standard deviations.

The online version of this article includes the following figure supplement(s) for figure 3:

**Figure supplement 1.** Select microbial and host metabolic parameters under wild colonization treatment.

## Diet versus host taxon effects on domesticated gut microbial composition in canids

To test if our findings were generalizable beyond mice, we conducted an analogous reciprocal diet experiment in captive sympatric populations of wolves and dogs (*Figure 4A*). We tracked gut microbial dynamics in these canids for 1 week on their standard diet (raw carcasses or commercial dog food, respectively) and 1 week on the reciprocal diet. As in the mouse experiment, we found that host taxon (wild or domesticated) explained the largest amount of variation in gut microbiota composition ($p<0.001$, $R^2 = 0.098$, $F = 13.730$, PERMANOVA), but that diet ($p<0.001$, $R^2 = 0.058$, $F = 8.151$) and a host taxon by diet interaction term ($p<0.001$, $R^2 = 0.028$, $F = 3.934$) were also significant (*Figure 4B*, *Figure 4—figure supplement 1*). There were significant differences among experimental groups in the magnitude of their ordination shifts along the first NMDS axis over the experimental periods ($p<0.001$, linear mixed effects model likelihood test; *Figure 4D*). As in the mouse experiments, we observed that animals on reciprocal diet treatments ($Dom_H/Wild_D$; $Wild_H/Dom_D$) moved significantly toward the habitual gut microbial profile of the other species ($p<0.05$, one-sample Wilcoxon tests; *Figure 4D*), while the microbiota of animals consuming their habitual diet ($Dom_H/Dom_D$; $Wild_H/Wild_D$) did not shift predictably ($p>0.100$).

In addition, we again observed an asymmetry between domesticated and wild animals in the degree to which the gut microbiota responded to diet. On experimental diets, dogs and wolves differed significantly in their dissimilarity to diet controls ($p<0.001$, Kruskal–Wallis test; *Figure 4E*), with the gut microbial communities of dogs fed raw carcasses resembling those of wolves at baseline but the gut microbial communities of wolves fed dog food remaining distinct from those of dogs at baseline ($p=0.001$, Mann–Whitney U test).

The difference in the direction of asymmetry between the canid and mouse experiments may be explained by the different trends in dietary ecology between carnivores and omnivores during domestication. Carnivores, through the addition of extensive carbohydrates to their diet (*Wolfe et al., 2007*), likely encounter more diverse diets in captivity than in the wild, whereas captive herbivores and omnivores typically eat a lesser number of plant species or are maintained on a single feed mix. Supporting this, we found that dogs initially had significantly higher OTU richness ($p<0.001$, *Figure 4—figure supplement 2*) and Shannon index ($p=0.003$, *Figure 4C*) than wolves, but that reciprocal diets led to a switch in diversity (richness: $p=0.002$, Mann–Whitney U tests), with wolves becoming more diverse when fed dog food while dogs lost diversity when fed raw carcasses (*Figure 4—figure supplement 2*).

## Analogous pressures in the human gut microbiota

We next explored the extent to which humans harbor gut microbial signatures analogous to those of domestication. Given evidence that the gut microbiota of domesticated animals is shaped by both ecology and speciation, we began by comparing humans to chimpanzees, one of our two closest living relatives. Humans may have undergone a form of self-domestication as a result of selection against aggression (*Wrangham, 2018*; *Theofanopoulou et al., 2017*) in addition to significant ecological change since our divergence from *Pan*, suggesting that the gut microbial signatures of animal domestication and *Pan–Homo* speciation could share features in common. We first compared samples that we collected from industrialized humans and wild chimpanzees, finding that the gut microbial communities of these humans and chimpanzees exhibited differences that paralleled those observed between domesticated animals and their wild counterparts when compared in the same ordination space ($p<0.001$, Mann–Whitney U test; *Figure 5A, B*). Microbial density ($p=0.002$, Mann–Whitney U test) and Shannon index ($p=0.018$; *Figure 1—figure supplement 4*) also differed between industrialized humans and wild chimpanzees, confirming prior reports that industrialized humans harbor microbial communities with substantially lower alpha-diversity (*Smits et al., 2017*). We found only a marginal difference in between-conspecific variability in the gut microbiota of industrialized humans and wild chimpanzees ($p=0.092$, $F = 3.0987$; permutation test for F). Including these human–chimpanzee comparisons in our analysis of the relationship between gut microbiota dissimilarity and the time since dyad divergence strengthened the observed relationship ($p<0.001$, $r = 0.5251$; Mantel test), with a conservative divergence time of 6.5 million years assumed for *Pan–Homo* in this analysis.

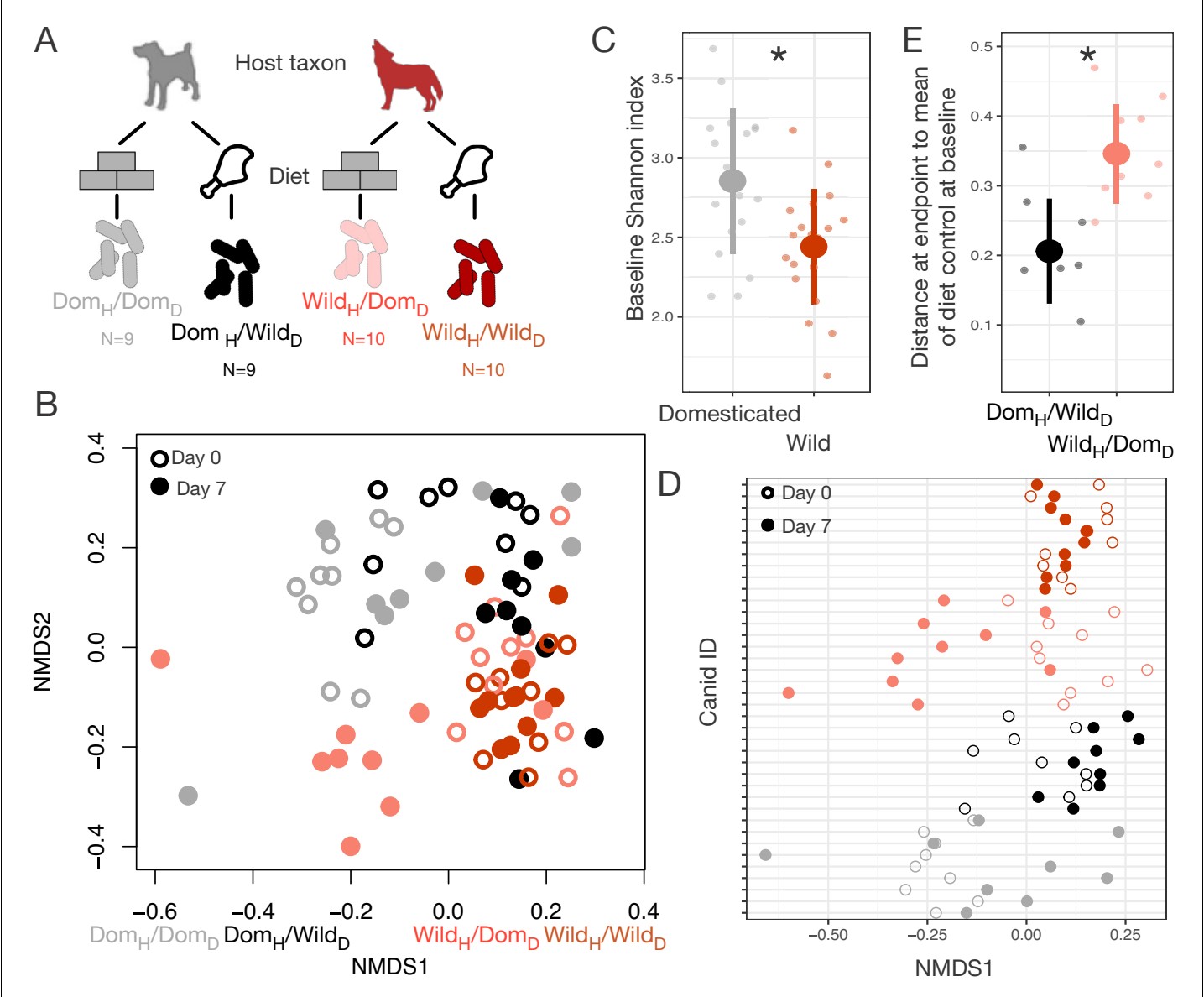

**Figure 4.** Microbial differences between wild and domesticated canids can be partially overcome by diet shifts. (A) Design scheme for fully factorial host taxon by diet canid experiment (N = 9 dogs or N = 10 wolves per diet group). (B) Nonmetric multidimensional scaling (NMDS) ordination of Bray–Curtis dissimilarities showing changes for canids from day 0 (open circle) to day 7 (filled circle) by experimental groups (color). Composition varied by host taxon (p<0.001, $R^2$ = 0.098, F = 13.70, permutational MANOVA), diet (p<0.001, $R^2$ = 0.058, F = 8.15), and a host taxon by diet interaction (p<0.001, $R^2$ = 0.028, F = 3.93). (C) Shannon index differed between dogs and wolves on day 0 (p=0.003, Mann–Whitney U test). (D) Canids on reciprocal diets ($Dom_H/Wild_D$ and $Wild_H/Dom_D$) but not control diets moved in opposite directions along Bray–Curtis ordination NMDS axis 1 over time (p=0.004 and 0.002, respectively, one-sample Wilcoxon tests). (E) At the end of the experiment, distance to the mean of diet controls at baseline ($Dom_H/Dom_D$ and $Wild_H/Wild_D$) was lower for dogs than for wolves on reciprocal diets (p=0.001, Mann–Whitney U test). * indicates p<0.05, Mann–Whitney U test. Large circles are means; bars show standard deviations.

The online version of this article includes the following figure supplement(s) for figure 4:

**Figure supplement 1.** Microbiota composition at all time points during the canid diet swap experiment.

**Figure supplement 2.** Select microbial parameters under canid diet swap.

However, given the vast ecological differences between wild chimpanzees and industrialized humans, it remained unclear the extent to which these *Pan–Homo* differences reflected host phylogenetic distance as opposed to ecology. To better gauge the divergence attributable to phylogenetic distance versus ecology, we proceeded to compare the gut microbial communities of humans

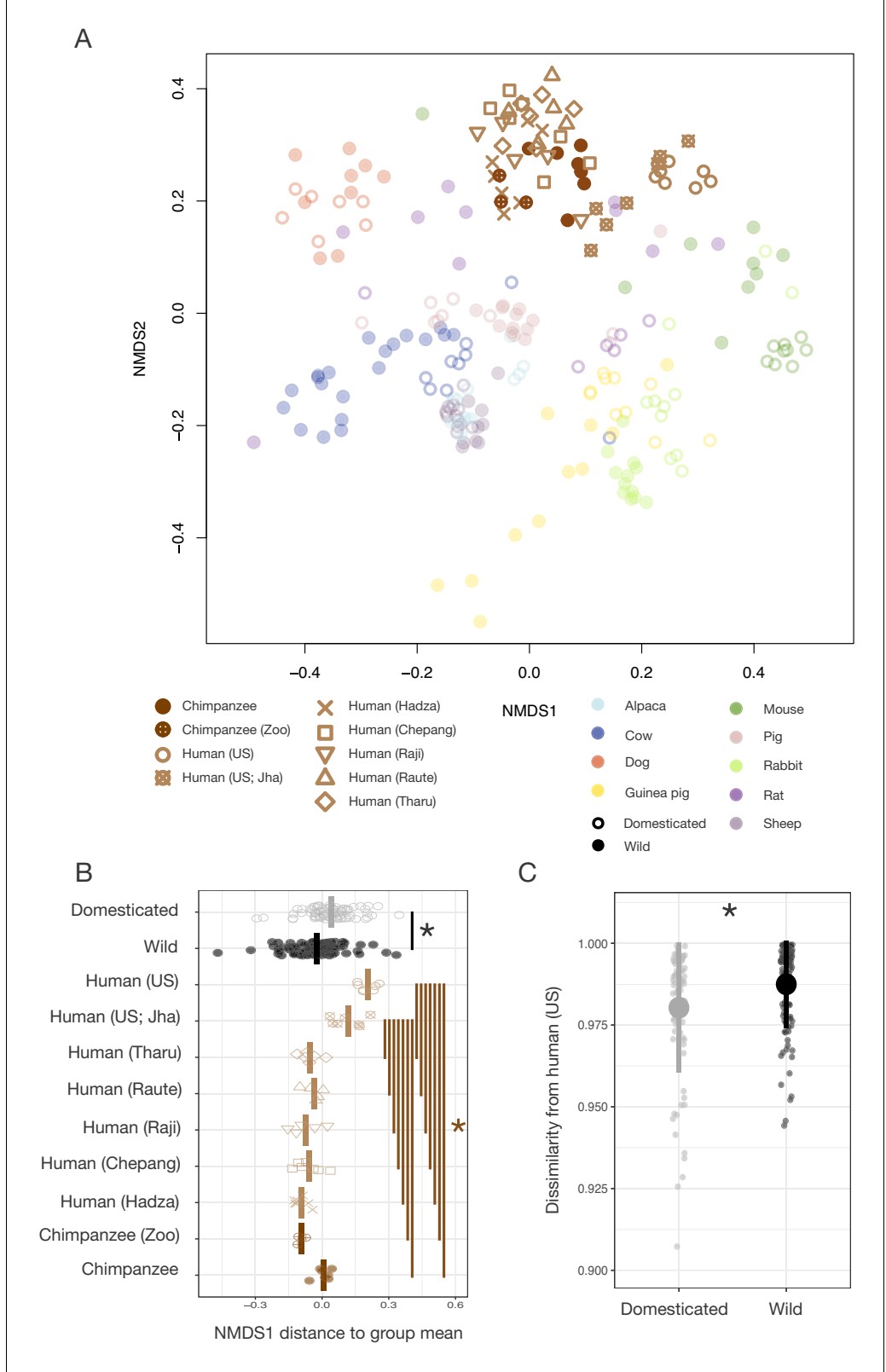

**Figure 5.** Differences in gut microbial communities between industrialized humans and wild chimpanzees parallel those observed between domesticated and wild mammals. (**A**) Nonmetric multidimensional scaling (NMDS) ordination of Bray–Curtis dissimilarities in the gut microbiota illustrates that industrialized human populations (US and US Jha) exhibit similar trends relative to wild chimpanzees as domesticated animals do to wild animals, but that non-industrialized human populations (Hadza, Chepang, Raji, Raute, and Tharu) do not (N = 5–7 individuals per primate population

*Figure 5 continued on next page*

*Figure 5 continued*

and 5–20 individuals per other animal species). (B) Distance along the first Bray–Curtis ordination NMDS axis relative to group mean differs in the same direction for the two industrialized human populations relative to wild chimpanzees or non-industrialized human populations as for domesticated animals relative to wild animals (p<0.05, Mann–Whitney U tests, N = 7–99). (C) The gut microbial communities of wild animals are more dissimilar to those of industrialized humans than are those of domesticated animals (p<0.001, bootstrapped Mann–Whitney U test, N = 82 domesticated and 99 wild). * indicates p<0.05, Mann–Whitney U test. Large shapes are means; bars in (C) show standard deviations.

The online version of this article includes the following figure supplement(s) for figure 5:

**Figure supplement 1.** Trends in gut microbial taxa previously linked to differing human lifestyles.

**Figure supplement 2.** Dissimilarity in the gut microbiota among chimpanzees and humans.

living in industrialized versus non-industrialized subsistence or agricultural societies, who are all equi-distantly related to chimpanzees. Reanalysis of our cross-species comparison including published data on human populations in rural Nepal and Tanzania pursuing various non-industrialized lifestyles *Jha et al., 2018* found that the gut microbial communities of these non-industrialized populations differed substantially from those of two independent U.S. samples, instead clustering more closely to those of chimpanzees in this ordination space (*Figure 5A*). Only the gut microbial communities of our industrialized populations show the rightward ordination shift along the Bray–Curtis NMDS axis 1 that is also exhibited by the gut microbial communities of domesticated animals (p<0.05, Mann–Whitney U tests; *Figure 5B*). Moreover, bacterial taxa previously found to distinguish among human lifestyles (*Smits et al., 2017*) typically had relative abundances that varied in the same direction between wild and domesticated animals as among wild chimpanzees, non-industrialized human populations, and industrialized human populations (*Figure 5—figure supplement 1*). These trends were clearest in the bacterial family Bacteroidaceae, which exhibited a continuous increase from chimpanzees to non-industrialized populations to industrialized populations as well as an increase in domesticated relative to wild animals (p=0.008, Mann–Whitney U test).

Together, these data indicate that the human gut microbiota does not carry a global signal of domestication, as would be predicted under a hypothesis of human gut microbial self-domestication. Rather, the corresponding gut microbial responses to domestication and industrialization suggest that these responses are more likely driven by common ecological factors, a conclusion further supported by the observation that the gut microbial communities of domesticated animals were more similar to those of industrialized humans than were those of their wild animal counterparts (p<0.001, bootstrapped Mann–Whitney U test; *Figure 5C*). Notably, the gut microbial communities of domesticated animals and industrialized humans most closely resembled one another for companion animals (p<0.001, Kruskal–Wallis test; *Figure 1—figure supplement 2*), presumably reflecting their greater ecological convergence and degree of physical contact (*Song et al., 2013*). Also supporting the role of ecology driving these trends, we found that an independent sample of captive chimpanzees did not cluster exclusively with the wild chimpanzee samples; indeed, their gut microbial communities were more similar to those of non-industrialized humans than to those of wild chimpanzees (p<0.001, Mann–Whitney U tests; *Figure 5—figure supplement 2*).

## Discussion

Our data demonstrate that while domestication has not led to a convergent 'domesticated micro-biota,' there is nevertheless a significant signal of domestication on gut community composition across diverse mammalian hosts. Furthermore, our experimental and cross-sectional analyses suggest that the domestication effect can, in large part, be ascribed to environmental rather than genetic changes. As in many comparative microbiota studies, host taxonomy was responsible for the largest component of variation in our cross-species analyses, but the contribution of domestication status was comparable to those of diet type and host physiology, factors previously identified as key drivers of the mammalian gut microbiota (*Ley et al., 2008*; *McKenzie et al., 2017*). Experimental diet intervention in wild/domesticated pairs reduced gut microbial dissimilarity and alpha-diversity differences between members of the dyad over short time scales. However, differences due to loss or gain of taxa during domestication could not be overcome by diet shifts alone, necessitating experimental recolonization. Together, these results indicate that domestication has played a large role in shaping the microbiota and, through husbandry practices, likely continues to do so today,

suggesting that studying variations in animal husbandry practices may illuminate new levers for manipulating the mammalian gut microbiota (*Velazquez et al., 2019*; *Villarino et al., 2016*; *Schmidt et al., 2019*).

Although there are many shared features of contemporary ecology and historic artificial selection on domestic animals, it is perhaps unsurprising that domestication has not produced a single, convergent domesticated gut microbiota. The animals we characterized represent a diverse set of lineages in the mammalian clade, and thus their microbiota also differ due to variation in factors such as gut structure and size, passage rate, diet, and biogeography (*Ley et al., 2008*; *Youngblut et al., 2019*). While the domestication signal was comparable in magnitude to those of gut physiology and diet type, it does not mask those fundamental structuring forces. Furthermore, the particulars of a domesticated lineage can help clarify what aspects of ecology are most salient to the domestication effect. Cases where domestication effects are weaker in our comparative study generally consist of animals where the ecological change associated with domestication has been small – for example, sheep and pigs, whose diets may be quite similar to their wild progenitors, at least when kept in the non-industrialized agricultural settings that were sampled (*McClure, 2013*) – or where ecological changes are in the opposite direction from the domesticated norm – for example, canids, where the domesticate diet typically involves lower protein and higher carbohydrate levels than wild diets, instead of the higher protein levels seen in most laboratory or farm animals (*Axelsson et al., 2013*).

Our reciprocal diet experiments in mice and canids substantiate our claim that ecology plays a predominant role in shaping the domesticated gut microbiota. However, they do not pinpoint the mechanism(s) for these effects. Variability in diet or other aspects of ecology and their concomitant effects on host physiology (e.g., passage rate) can alter microbial composition or abundance through changes in the selective landscape that microbes experience and changes in environmental exposure (*David et al., 2014*; *Carmody et al., 2019*). Animals may experience altered bacterial colonization, leading directly to changes in composition, and/or viral colonization, which could then alter the bacterial community if new bacteriophages target gut bacteria or if eukaryotic viruses activate the host immune system, leading to transformations in the gut environment. That the gut microbial impacts of change in a single ecological variable like diet were sufficiently profound to balance those of host taxon identity suggests that suites of ecological variables changing together, such as during domestication or industrialization, may have collectively exerted an even larger influence (*Jha et al., 2018*).

Of course, microbiota changes were not the only pathway for animals undergoing domestication to respond to changing ecological factors. For example, genetic changes have enhanced the capacity for starch digestion in dogs (*Axelsson et al., 2013*; *Reiter et al., 2016*). Nevertheless, the increased microbial diversity and shifts in microbial composition that we observed in dogs may have additionally contributed to carbohydrate digestion. Indeed, dogs fed conventional diets have greater representation of carbohydrate metabolism genes in their gut metagenomes than do dogs fed meat-based diets (*Alessandri et al., 2019*). Notably, the microbiota has been found to supplement evolutionary responses during dietary niche expansion in wild animals that consume plants high in toxins (*Kohl et al., 2014*). As such, although hosts and their various gut microbial taxa are each expected to pursue their own fitness interests, gut microbial disparities observed between domesticated and wild animals, and more generally in other organisms under rapid environmental change, could potentially be adaptive for the host (*Alberdi et al., 2016*).

Beyond host support of a gut microbiota that can better digest a domesticate diet, humans may have selected for animals harboring a microbiota that helped them grow and reproduce well on such diets, thereby applying unconscious selection on the microbiota (*Zohary et al., 1998*). Changes in microbial function that enhanced host dietary energy harvest, survivorship, or reproduction may have been particularly important early in domestication, before host evolution occurred, although that hypothesis remains to be tested empirically. Regardless, specialization of microbial performance on domesticate diets could conceivably have come at the cost of broader digestive capacity, as seen in the laboratory mouse microbiota, which was better at harvesting energy from domesticate diets than from wild diets (*Figure 2—figure supplement 2*). It may also have impacted the immunological functions of the gut microbiota. The elevated pathogen abundances found in wild populations overall may largely be ascribed to low pathogen abundances in laboratory animals (*Figure 1—figure supplement 2*), which are maintained under specific pathogen-free conditions that minimize the likelihood of infection. Under natural conditions, though, the domesticated microbiota may exhibit

reduced colonization resistance or immune system functioning (*Rosshart et al., 2017*; *Beura et al., 2016*; *Rosshart et al., 2019*), resulting in higher pathogen colonization, as observed here in agricultural animals. Future studies examining the trade-offs among microbially mediated functions, like digestive capacity, reproduction, and immunity, will help to illuminate the complex selection pressures shaping domesticated animals and their gut microbiota (*Reese and Kearney, 2019*).

We observed some correspondence between the gut microbial signatures of animal domestication and human industrialization that is most likely attributable to convergent ecological changes. The observation that gut microbial divergence among *Pan* and *Homo* primarily affects industrialized populations specifically implicates recent ecological changes as opposed to either ecological changes with deeper roots in human evolution or host evolutionary changes. Many recent human ecological changes involve accelerations of basic patterns established during the evolution of *Homo*, including increased proportion of calories from fat and protein, increased dependence on animal source foods, and extensive food processing involving both chemical and physical changes to food (*Carmody, 2017*). Other ecological changes are likely specific to industrialization, including reduced physical activity, high population density, and antibiotic use. These factors would be absent even in populations currently transitioning from subsistence to industrialized lifestyles (*Jha et al., 2018*), but may overlap with changes experienced by domesticated animals in their diets, habitats, and social milieu. While we limited our analysis to human–chimpanzee comparisons because *Pan* is the closest sister clade to *Homo*, recent work has indicated that the human gut microbiota is more similar to that of baboons (*Gomez et al., 2019*; *Amato et al., 2019*). Baboons are more distantly related to humans but have been argued to be closer in terms of diet and dietary physiology (*Codron et al., 2008*; *Lambert, 1998*), accentuating our finding of the importance of ecological factors in shaping the microbiota. Further work will be required to assess the specific combination of ecological factors driving similarities between domesticated and industrialized gut microbial signatures.

Because laboratory animals demonstrate some of the largest overall differences relative to their wild counterparts, they might be expected to have high translational potential as models for studying the gut microbiota of industrialized human populations. However, recent findings show that laboratory mice are poorer immunological models for humans in industrialized settings than are wild mice or laboratory mice harboring a wild microbiota (*Beura et al., 2016*; *Rosshart et al., 2019*). While the industrialized human gut microbiota exhibits parallels to those of domesticated animals, it may experience a broader array of environments and greater temporal variability; for example, greater ecological variability may explain the elevated gut microbial Shannon diversity seen in humans as compared to laboratory animals (*Figure 1—figure supplement 2*). Alternatively, it may be that domesticated laboratory animals are strong models for some aspects of host–microbe biology other than immunology. Certainly, studies of non-domesticated animals will be necessary to understand the natural history of host–microbe interactions (*Reese and Kearney, 2019*; *Hird, 2017*), as well as to determine the most appropriate models for translational research.

The fact that laboratory mice were permissive of recolonization by wild strains indicates that the local extinctions that occurred during domestication and/or generations in captivity can potentially be mitigated, thereby potentially improving the utility of these animals for research. Previous work has relied on laboratory mice colonized with a wild microbiota but fed standard laboratory chow (*Rosshart et al., 2017*; *Rosshart et al., 2019*) or on wild mice fed wild diets (*Martínez-Mota et al., 2020*). A combination of these approaches – adding wild gut microbial community members and feeding wild diet – would be expected to best support a wild gut microbiota in laboratory mice. A wild-microbiota laboratory-genotype model could be especially useful for studying infection challenges, disentangling host gene versus microbiota contributions to disease phenotypes, and testing for host–microbiota coevolution (*Rosshart et al., 2019*).

More generally, our data add to growing evidence that the gut microbiota is finely tuned to variations in the environment, affording at once expanded opportunities for biological mismatch to arise between the host and microbiota and for rapid microbiota-mediated host adaptation to novel environments. Further work to characterize the ecological significance of gut microbial plasticity will help reveal the fundamental nature of the host–microbial relationship, the conditions under which plasticity is beneficial versus detrimental, and the ecological conditions promoting cooperative, commensal, and competitive dynamics. The answers will have profound implications for our definition and pursuit of a healthy gut microbiome.

# Materials and methods

## Key resources table

| Reagent type (species) or resource | Designation | Source or reference | Identifiers | Additional information |
|---|---|---|---|---|
| Biological sample (*Bos taurus*) | Feces | This paper | | N = 9, sex unknown |
| Biological sample (*Bison bison*) | Feces | This paper | | N = 20, sex unknown |
| Biological sample (*Ovis aries*) | Feces | This paper | | N = 13, twelve females |
| Biological sample (*Ovis canadensis*) | Feces | This paper | | N = 10, sex unknown |
| Biological sample (*Sus scrofa domesticus*) | Feces | This paper | | N = 9, sex unknown |
| Biological sample (*Sus scrofa*) | Feces | This paper | | N = 16, five females |
| Biological sample (*Vicugna pacos*) | Feces | This paper | | N = 8, sex unknown |
| Biological sample (*Vicugna vicugna*) | Feces | This paper | | N = 5, two females |
| Biological sample (*Canis lupus familiaris*) | Feces | This paper | | Comparative: N = 7, four females Experiment: N = 9, sex unknown |
| Biological sample (*Canis lupus*) | Feces | This paper | | Comparative: N = 9, sex unknown Experiment: N = 10, sex unknown |
| Biological sample (*Oryctolagus cuniculus*) | Feces | This paper | | Domesticated: N = 11, four females Wild: N = 12, sex unknown |
| Biological sample (*Cavia porcellus*) | Feces | This paper | | N = 10, zero female |
| Biological sample (*Cavia tschudii*) | Feces | This paper | | N = 11, sex unknown |
| Biological sample (*Mus musculus*) | Feces | This paper | | Comparative: N = 9 (domesticated), zero female N = 9 (wild), sex unknown Experiments: N = 49 (domesticated), zero female N = 6 (wild), sex unknown |
| Biological sample (*Rattus norvegicus*) | Feces | This paper | | Domesticated: N = 6, sex unknown |

*Continued on next page*

*Continued*

| Reagent type (species) or resource | Designation | Source or reference | Identifiers | Additional information |
|---|---|---|---|---|
| Biological sample (*Rattus norvegicus*) | Intestinal sample | This paper | | Wild: N = 10, three females |
| Biological sample (*Pan troglodytes*) | Feces | This paper | | Wild: N = 7, seven females<br>Captive: N = 3, two females |
| Biological sample (*Homo sapiens*) | Feces | This paper | | N = 7, five females |
| Sequence-based reagent | 515F | *Caporaso et al., 2011* | PCR primers | GTGCCAGCMGCCGCGGTAA |
| Sequenced-based reagent | 806R | *Caporaso et al., 2012* | PCR primers | GGACTACNVGGGTWTCTAAT |
| Software, algorithm | R | R Core Team | Version 3.3 | |
| Software, algorithm | QIIME | *Caporaso et al., 2010* | Version 1.8 | |
| Software, algorithm | vegan | *Oksanen et al., 2017* | | |
| Software, algorithm | lme4 | *Bates et al., 2015* | | |
| Software, algorithm | TimeTree | *Kumar et al., 2017* | | http://timetree.org |
| Software, algorithm | boot | *Canty and Ripley, 2020* | Version 1.3-25 | |

## Fecal sample collection

Distal gut microbiota samples from a range of non-human species were collected by authors or collaborators. Fecal samples from non-human mammals were collected from the ground within seconds to a few hours (<6) of production over the course of 2017 and 2018. In the case of artiodactyl, carnivore, lagomorph, and rodent feces, this approach precluded the need for institutional approval. Wild chimpanzee fecal samples were collected by field assistants under the approval of the University of New Mexico IACUC (protocol 18-200739-MC) and with permission of the Uganda Wildlife Authority and Uganda National Council for Science and Technology. Captive chimpanzee fecal samples were collected passively by keepers at Southwick's Zoo, Mendon, MA. Human samples were self-collected by healthy study participants after providing written informed consent under the approval of the Harvard University IRB (protocol 17-1016) (*Carmody et al., 2019*). Samples were immediately frozen prior to permanent storage at –80˚C. The only exceptions were wild vicuña and wild chimpanzee samples, which were preserved in RNAlater stabilization solution (Invitrogen) due to logistical issues in transportation from remote sampling locales. RNAlater was removed from these samples with centrifugation prior to further processing, and while sample preservation method was significantly associated with microbiota composition (p<0.001, PERMANOVA) it explained only a minor portion ($R^2$ = 0.01) of the variation in beta-diversity. Sample sizes were chosen based on animal availability with a N > 5 for all species.

## Domesticated animals

Domesticated sheep (*Ovis aries*; N = 11, ten females), cattle (*Bos taurus*; N = 9, sex unknown), and pig (*Sus scrofa domesticus*; N = 9, sex unknown) fecal samples were collected from a farm in Vershire, Vermont (VT). Domesticated alpaca (*Vicugna pacos*; N = 8, sex unknown) and domesticated sheep (*O. aries*; N = 2, two females) fecal samples were collected from a farm in Groton, Massachusetts (MA1). Mouse (*Mus musculus*, N = 9, zero female), rat (*Rattus norvegicus*; N = 6, sex unknown),

and guinea pig (*Cavia porcellus*; N = 10, zero female) fecal samples were collected from animals in Harvard laboratory facilities (MA2). Domesticated rabbit (*Oryctolagus cuniculus*; N = 11, four females) fecal samples were collected from a shelter in Billerica, Massachusetts (MA3). Dog (*Canis lupus familiaris*; N = 7, four females) fecal samples were collected from personal pets in Stacy, Minnesota (MN). All samples were collected in summer or fall 2017. We have limited ability to distinguish between locale and species effects since all but one species (sheep) had samples collected from only one locale. Some locales had multiple species present, and we do find a significant effect of locale on overall microbial community composition even when correcting for host phylogeny effects (p<0.001, $R^2$ = 0.16, F = 6.14, PERMANOVA). However, it is clear that locale does not necessarily lead to convergent microbiota across taxa as evidenced by the low clustering by site in NMDS ordination space (*Figure 1—figure supplement 1*). When analyzing just the sheep samples, we find a minimal effect of locale (p=0.023, $R^2$ = 0.07, F = 2.08, PERMANOVA).

Time since domestication for each species pair was drawn from published data (*Zeder, 2012*; *Morand et al., 2014*; *Driscoll et al., 2009*; *Goñalons and Yacobaccio, 2006*). Time since divergence from wild progenitor or paired wild sample was taken from http://timetree.org (*Kumar et al., 2017*) because not all species in our sample set had existing genome assemblies of sufficient quality from which to infer a genome-based phylogeny. For animals with the same species name as their wild pair (e.g., *Sus scrofa* for both pigs and boars), which TimeTree treats as the same node, we used time since domestication estimates in lieu of time since divergence. In two cases (alpaca/vicuña and guinea pigs), the species have different names, enabling TreeTime to estimate time since divergence. Notably, these estimates are much larger than the time since domestication estimates for those dyads despite the fact that the wild species sampled are widely considered to be the progenitors of the domesticates. We include analyses with both time since divergence and time since domestication, considering the former to be more conservative estimates of relatedness. Gut physiology and diet classifications were assigned based on published literature (*Stevens and Hume, 2004*) and are listed in *Supplementary file 2*.

## Wild animals

Wild boar (*Sus scrofa*; N = 16, five females) fecal samples were collected from adults and juveniles in southeastern Alabama (AL) during fall 2017. Rat (*Rattus norvegicus*; N = 10, three females) distal gut samples from adults and juveniles were collected directly from the colon shortly following trapping in New York City (NY) between February and May 2017 (*Combs et al., 2018*). Bison (*Bison bison*, N = 20, sex unknown) fecal samples were collected from a semi-free-ranging population in Elk Island National Park, Alberta, Canada (CAN) during 2013 (*Weese et al., 2014*). Wild house mouse (*Mus musculus*, N = 9, sex unknown) fecal samples were collected from live-trapped animals in the Cambridge, Massachusetts area (MA4) during winter 2018. Pursuant to Massachusetts state law, permits were not necessary to trap animals indoors. Wild European rabbit (*Oryctolagus cuniculus*; N = 12, sex unknown) fecal samples were collected in Mértola, Portugal (POR), during spring 2018. Bighorn sheep (*Ovis canadensis*; N = 10, sex unknown) fecal samples were collected during 2017 and 2018 in Wyoming (WY). Vicuña (*Vicugna vicugna*; N = 5, two females) fecal samples were collected during spring 2018 from a captive population in Santiago, Chile (CHL) that was free-grazing but supplemented with hay. Wild guinea pig (*Cavia tschudii*, N = 11, sex unknown) fecal samples were collected at a facility in Lima, Peru, (PER) during spring 2018. Wolf (*Canis lupus*; N = 9, sex unknown) fecal samples were collected during fall 2017 from captive packs fed an exclusively raw diet at the Wildlife Science Center (WSC) sanctuary in Stacy, Minnesota (MN). Wild chimpanzee (*Pan troglodytes schweinfurthii*, N = 7, seven females) fecal samples were collected between September 2015 and January 2016 from adult members of the Kanyawara community in the Kibale National Park, Uganda; samples were randomly selected from adults from a larger set initially prepared for a separate project (*Reese et al., 2021*). Captive chimpanzee (N = 3, two females) fecal samples were collected in May 2019 from adults at Southwick's Zoo in Mendon, MA.

## Humans

Fecal samples were collected from healthy adult humans (N = 7, five females) residing in the Cambridge, Massachusetts area. All participants were provided with sterile study kits and self-collected fecal samples during the same 3-day period in December 2017. During this period, participants

freely consumed their habitual diets. Fecal samples were immediately stored at –20˚C and were transferred within 24 hr to permanent storage at –80˚C.

## Animal experiments

### Wild mouse capture

*Mus musculus* were introduced to North America from Western Europe and are now commonly found in commensal settings (*Schwarz and Schwarz, 1943*). We set out Sherman live traps in the evenings in buildings and barns during February 2018. Traps were baited with peanut butter and a small cube of apple and outfitted with sufficient bedding and food to sustain an adult mouse for at least 48 hr. They were checked the following morning to minimize time spent in the traps. Rodents were transferred from their traps to a plastic bag, and unwanted rodent species were released immediately. Mice that were identified as *M. musculus* (rather than *Peromyscus* spp., also common in Massachusetts) were transferred to temporary cages for transport to lab facilities. At time of capture, we collected fecal samples and body swabs for zoonoses testing by Charles River. All individuals were tested for fur mites; MAV 1 and 2; MHV; MPV/MVM; MRV; mousepox; POLY; REO; LCMV; LDV; TMEV/GDVII; SEND; PVM; *Mycoplasma; Mycoplasma pulmonis; Filobacterium rodentium* (formerly CAR Bacillus); *Citrobacter rodentium; Clostridium piliforme; Corynebacterium kutscheri; Corynebacterium bovis; Streptobacillus moniliformis*; and pinworm. The only agent of concern found was fur mites. Because animals were not treated for parasites or pathogens in order to preserve their wild gut microbiota signature, they were housed under non-specific pathogen free (SPF) conditions at Harvard's Concord Field Station. Mice were allowed at least 3 days to adjust to laboratory conditions without handling and provided with a wild mouse diet (a mix of bird seed [Wagner's Eastern Regional Blend Deluxe Wild Bird Food] and Bio-Serv freeze-dried mealworms; *Supplementary file 3)* before the beginning of the experiment. All mice were housed singly from the time of arrival at the Concord Field Station and had access to water and food ad libitum.

### Wild/laboratory mice reciprocal diet experiment

A total of 10 wild mice were captured for this experiment. Of these, two were deemed too young for inclusion in the study, one died before beginning the experiment, and one died during the course of the experiment. As a result, we collected six wild mice (Wild$_H$) that were included in the full study. In addition to the wild mice, male C57BL/6 mice 10–12 weeks of age with a conventional microbiota were purchased from Charles River Laboratories for inclusion in the study (Dom$_H$). All mouse experiments were conducted in accordance with the National Institutes of Health Guide for the Care and Use of Laboratory Animals using protocols approved by the Harvard University Institutional Animal Care and Use Committee (protocol 17-11-315). The sample size for the laboratory animal group was chosen following power analyses to allow for β less than 0.1; the sample size for the wild animal group was chosen based on animal availability (N = 3 per diet treatment). The experiment was conducted once. All mice were housed singly from the time of arrival at the Concord Field Station and had access to water and food ad libitum. Mice were provided nesting material and plastic enrichment housing atop corncob bedding. The mice were maintained in a 20–22˚C room with natural light cycles.

Mice, both wild and laboratory, were randomly assigned to one of two dietary treatment groups (N = 10 laboratory mice or three wild mice per group). The first group (domesticate diet: Dom$_D$) was provided ad libitum mouse chow (Prolab Isopro RMH 3000) in overhead food hoppers, as is standard in mouse studies. The second group (wild diet: Wild$_D$) was provided a mix of bird seed (Wagner's Eastern Regional Blend Deluxe Wild Bird Food) and freeze-dried mealworms (*Supplementary file 3*) in excess of predicted consumption. The food was placed in the corncob bedding to simulate foraging.

Before initiating the dietary interventions, all individuals were weighed and multiple fecal samples were collected. The mice were then returned to a new, clean cage with the treatment diet present. Over the next week, fecal samples and weights were collected daily for each mouse. The amount of food remaining was weighed and additional wild diet was added daily. One week after beginning the experiment, mice were weighed and fecal samples were collected, then mice were moved to clean cages. Weights and fecal samples were henceforth collected weekly (days 14, 21, and 28) until the end of the experiment, although additional food was added biweekly for individuals assigned to

the wild diet treatment. Additional chow was added to hoppers for individuals assigned to the conventional diet treatment, and all water bottles were refilled as necessary. At the end of each week, food hoppers were weighed ($Dom_D$), and cage bedding was collected and sifted to quantify uneaten food ($Wild_D$), determine total weekly fecal production (all groups during week 3), as well as to provide fecal samples for bomb calorimetry (6050 Calorimeter, Parr). All calorimetry results were adjusted for the average weekly fecal production and average weekly food intake of each experimental group. At the end of the experiment (days 28–30), mice were humanely sacrificed via $CO_2$ euthanasia.

## Wild/laboratory mice gavage experiment

Thirty 10-week-old male C57BL/6 mice with a conventional microbiota were purchased from Charles River Laboratories for inclusion in the study. The sample size was chosen following power analyses to allow for β less than 0.1; the experiment was conducted once. Mice were cohoused in litter groups of 3–4 until beginning the study. Cage groups were spread across the treatment groups, with individuals randomly assigned to a diet and colonization treatment. There were three treatment groups: wild colonized/wild diet ($Wild_C$/$Wild_D$); wild colonized/domesticate diet ($Wild_C$/$Dom_D$); or phosphate-buffered saline (PBS) gavage/wild diet ($PBS_C$/$Wild_D$). The latter served as a colonization control, emulating the $Dom_H$/$Wild_D$ group from the reciprocal diet mouse experiment.

On the first day of study, fecal samples were collected from each mouse and the mice were weighed before colonization. Mice receiving a wild microbiota were colonized with cecal contents collected from one randomly selected $Wild_H$/$Wild_D$ individual in the wild/laboratory experiment (see above). The cecal contents were prepared following *Rosshart et al., 2017*. In short, frozen cecal contents were resuspended in sterile-reduced PBS (1:1 g:ml) under anaerobic conditions then diluted 1:30. Each recipient mouse received a single dose of 100 to 150 μl cecal solution via oral gavage. PBS control mice received 100–150 μl sterile-reduced PBS via oral gavage.

Following gavage, mice were transferred to single housing in new, clean cages with the treatment diet present. Mice receiving domesticate diet were provided ad libitum mouse chow (Prolab Isopro RMH 3000) in overhead food hoppers. Wild mouse diet consisted of a mix of bird seed (Wagner's Eastern Regional Blend Deluxe Wild Bird Food) and freeze-dried mealworms (*Supplementary file 3*), which was provided in excess of predicted consumption and placed in the corncob bedding to simulate foraging. All mice were provided with nesting material and plastic enrichment housing atop corncob bedding.

Additional fecal samples and weights were collected on days 1, 2, and 8 following gavage. After weights and fecal samples were collected on day 8, mice were humanely sacrificed via $CO_2$ euthanasia. During the course of the experiment, one mouse in the $PBS_C$/$Wild_D$ (control) treatment group died, resulting in a N = 9 for that group. At the end of the experiment, food hoppers were weighed ($Dom_D$), and cage bedding was collected and sifted to quantify uneaten food ($Wild_D$) and total weekly fecal production (all groups).

## Wolf/dog reciprocal diet experiment

Ten wolves (*Canis lupus*) and nine dogs (*Canis lupus familiaris*) participated in the study. The sample size for the canid experiment laboratory animal group was chosen following power analyses to allow for β less than 0.1; the experiment was conducted once. Wild-caught or captive-born wolves lived in packs of 2–6 wolves at the WSC (Stacy, MN). They were exposed to natural light cycles and weather conditions, with access to shelters and wolf-dug dens in their enclosures. Wolves had ad libitum access to water. Dogs enrolled in this study were privately owned in Stacy, MN, and were recruited to participate through their owners. Dogs were kept in their typical environment throughout the experiment. All canid experimentation was approved by the WSC IACUC (protocol HAR-001). Wolves were enrolled in the study from December 5–20, 2018, and dogs were enrolled from December 24, 2018 to January 8, 2019.

On every day of the study, across both the control and reciprocal diet arms, wolves were given inert colored glass beads via treats (~15 g raw meatballs). The beads can be passed naturally without harm to the animal and allowed for source identification for fecal samples in cohoused animals. Fecal samples were collected daily in a sterile manner then moved to −20°C storage before long-term storage at −80°C. For the first week of the experiment, all animals received a control diet that

matched their genetic background (*Supplementary file 3*) – raw chicken parts (4 lbs/animal) for wolves (Wild$_H$/Wild$_D$) and commercial dog food (Nutrisource Lamb Meal and Peas Recipe, Grain Free) for dogs (Dom$_H$/Dom$_D$). Fecal samples were collected at least once daily from wolf enclosures and the dogs' home environments without handling the animals. On day 8, wolves were provided no new food, but were able to complete consumption of previously provided food. Fecal samples collected on this day were considered baseline samples for the next arm of the experiment. Beginning on day 8, a week of reciprocal diet feeding was commenced. During this period, wolves were fed commercial dog food (Wild$_H$/Dom$_D$) and dogs were fed raw chicken parts (Dom$_H$/Wild$_D$). Daily fecal samples were again collected. Following completion of the study, animals were returned to their standard diet.

## Human sample meta-analysis

To compare the microbial differences observed between wild and domesticated animals and between humans and chimpanzees with differences linked specifically to industrialization, we also performed analyses including all of the samples outlined above plus a subset of published data from Jha and colleagues (*Jha et al., 2018*). To match sample sizes used in our human–chimpanzee contrast, we subsampled seven adults from their Chepang (Nepalese foragers), Raji (Nepalese foragers transitioning to subsistence farming), Raute (Nepalese foragers transitioning to subsistence farming), Tharu (Nepalese subsistence farmers), and American populations, as well as seven adults from the Hadza (Tanzanian hunter gatherers) population they analyze, which were originally described in another study (*Smits et al., 2017*). All data were downloaded from the European Nucleotide Archive. These populations represent extremes of industrialized and non-industrialized human lifestyles with the variation among the non-industrialized groups not covering the full breadth of intermediate lifestyles (e.g., modern agricultural or recent urban transplants). We believe that these extremes enable us to test how the human gut microbial communities respond to major ecological change of a magnitude that could be argued to approximate that experienced by gut microbial communities of animals undergoing domestication.

These samples were not necessarily collected or processed in an identical manner to each other or to the new data collected in this paper – namely, the Chepang, Raji, Raute, and Tharu samples were collected and preserved using OMNIgene kits while the American and Hadza samples were frozen, while all samples were extracted with the MoBio PowerSoil kit and sequenced on an Illumina MiSeq. Unfortunately, no existing published data on non-industrialized populations have been generated using exactly the same methods employed here. However, we reprocessed the sequences using the 16S rRNA gene amplicon QIIME pipeline described below and rarefied all samples to 10,000 reads depth to make the data as comparable as possible. Importantly, to the extent that these discrepancies introduce biases to our analyses, we expect they would do so in a manner agnostic to the comparison with our chimpanzee and American human samples. The high similarity between the US samples that we collected and those collected by Jha and colleagues supports this expectation. Furthermore, the fact that the non-industrialized Hadza samples were not stored with OMNIgene kits precludes conflating any non-industrialized signal with a sample-processing signal.

Specific taxa chosen for targeted analyses were identified from the human lifestyle analyses by Smits and colleagues (*Smits et al., 2017*); only taxa that had a non-zero abundance in wild chimpanzees were analyzed here. Time since *Pan–Homo* divergence was drawn from http://timetree.org (*Kumar et al., 2017*) to be consistent with domestication analyses.

## 16S rRNA gene analysis

### Extraction

Following collection during observational or experimental animal work, fecal samples were temporarily stored at −20℃ then moved to −80℃ for long-term storage. Individual mouse pellets or approximately 50 mg feces were used for DNA extraction using the E.Z.N.A. Soil DNA Kit (Omega, Norcross, GA) following manufacturer's instructions.

### Sequencing

We performed 16S rRNA gene amplicon sequencing on fecal samples to determine gut microbial community structure. We used custom barcoded primers (*Caporaso et al., 2011*) targeting the 515F

to 806Rb region of the 16S rRNA gene following published protocols (*Caporaso et al., 2011*; *Caporaso et al., 2012*; *Maurice et al., 2013*). Sequencing was conducted on an Illumina HiSeq 2500 with single-end 150 bp reads in the Bauer Core Facility at Harvard University. Sequence files were processed using QIIME version 1.8 (*Caporaso et al., 2010*). We demultiplexed the sequences using split_libraries_fastq.py, then used parallel_pick_otus_uclust_ref.py to carry out closed reference operational taxonomic unit (OTU) picking with 97% similarity using the default parameters under UCLUST. Microbial taxonomy for these OTUs was assigned in reference to the GreenGenes database (version 13.5) (*DeSantis et al., 2006*). We obtained $158,611 \pm 109,567$ assigned reads per sample.

## qPCR

To estimate total bacterial density, qPCR was performed on fecal DNA using the same primers as used for sequencing. qPCR assays were run using PerfeCTa SYBR Green SuperMix Reaction Mix (QuantaBio, Beverly, MA) on a Bio-Rad CFX384 Touch (Bio-Rad, Hercules, CA) in the Bauer Core Facility at Harvard University. Cycle-threshold values were standardized against a dilution curve of known concentration and then adjusted for the weight of fecal matter extracted.

## Statistical analyses

All statistical analyses were carried out in R (R Core Team, version 3.3). Alpha-diversity (Shannon index, OTU richness) were calculated for rarefied OTU tables (rarefaction limit of 17,500 for cross-species dataset; 27,000 for wild mouse study; 15,500 for the mouse colonization study; 7,500 for canid experiment). Beta-diversity (Bray–Curtis, Weighted UniFrac, Unweighted UniFrac) metrics were calculated using the vegan package (*Oksanen et al., 2017*) or QIIME on unrarefied data. All statistical tests performed were non-parametric. PERMANOVA was carried out with the adonis2 function in vegan with the domestication status variable nested within the species pair to correct for known relationships within dyads. To test how beta-diversity varied based on relatedness (within species, between wild–domesticated pairs, or among unrelated pairs), domestication type (relative to US human samples), or human/primate population (relative to zoo chimpanzee samples), we used a bootstrapping approach, thus correcting for the non-independence of dissimilarity measurements that include the same individuals in multiple comparisons. In short, we permuted the Mann-Whitney U test statistics and p values, resampling (25,000 permutations) with stratification specified by individual identity, using the boot package (*Canty and Ripley, 2020*). Variability in a species' microbial community composition was calculated with the permutest and betadisper functions in vegan. For changes in family-level abundance, a Bonferroni correction for multiple hypothesis correction was then applied to all test results.

Potential human pathogens were identified following published methods (*Kembel et al., 2012*; *Reese et al., 2016*). In short, we obtained a list of potential human pathogens, compiled by Kembel and colleagues (*Kembel et al., 2012*), then manually compared that list to the taxa identified to the genus or species level in our analysis. A subset of the data containing only these species was then analyzed for diversity with the same methods used for the total dataset.

To determine the consistency of gut microbial differences across ordination space due to domestication, *Pan–Homo* divergence, or industrialization in the observational study, we calculated the average position of the host dyad (e.g., pig/boar) or all primates (humans and chimpanzees) for axis 1 of the NMDS, then measured the displacement along each axis for an individual sample relative to that mean position. We tested for differences in these ordination shifts by domestication status or primate host taxonomy (e.g., chimpanzee versus US human). To estimate the direction and magnitude of changes in beta-diversity during the experimental studies, we tested whether inclusion of a treatment group term improved the performance of a linear mixed effects model relative to a model with only time and animal ID terms for predicting the NMDS1 axis value for an individual. These analyses allowed us to consider the direction of beta-diversity changes in addition to the magnitude. We estimated the direction and magnitude of dissimilarity from the expected community composition (donor microbial community in gavage experiment; baseline species average for $Dom_H/Dom_D$ or $Wild_H/Wild_D$ in diet experiments) as the length of the vector through the first axis of ordination space. In analyzing the experimental diet study data, we used the *lmer* and *anova* functions in the package lme4 (*Bates et al., 2015*) to perform likelihood tests comparing a linear mixed effects

model that included the variable of interest (i.e., treatment group) to a model that included only time variables. In both models, individual identity was included as random effects.

We explored the role of relatedness in structuring the cross-species dataset by (i) performing a Mantel test to compare divergence times and Bray–Curtis dissimilarities; (ii) testing for Spearman correlations between the NMDS shifts and the time since divergence and performing likelihood tests to compare a linear mixed effects model that included both domestication status and dyad as fixed effects and divergence time as a random effect with a model that only included the dyad and divergence time terms; and (iii) testing for Spearman correlations between the average dissimilarity within a wild–domesticated dyad (e.g., the average dissimilarity for all combinations of boar–pig pairs) and the time since domestication and time since divergence. We also used Mann–Whitney U tests to determine if dissimilarity between unrelated pairs was higher than for wild–domesticated dyads or within sets of conspecifics.

## Acknowledgements

We thank many collaborators for help in collecting wild and domesticated animal fecal samples, including Gwynne Durham and The Mountain School (cattle, pigs, sheep); Luina Greine Farm (alpaca); Steve Ditchkoff (wild boar); Jason Munshi-South and Matthew Combs (rats); Pedro Monterroso, Marisa Rodrigues, and Marketa Zimova (European rabbits); Margaret Gruen and Kyle Smith (dogs, wolves); Kevin Monteith (Bighorn sheep); J Scott Weese (bison); Cristián Bonacic (vicuña); Bridget Alex, Hopi Hoekstra, Nicholas Holowka, Irene Li, Daniel Lieberman, Mark Omura, and Antonia Prescott (wild mice). For help in collecting and processing wild chimpanzee samples, we thank Tony Goldberg, Zarin Machanda, Martin Muller, Emily Otali, Leah Owens, Sarah Phillips-Garcia, Richard Wrangham, and the staff of Kibale Chimpanzee Project. For assistance in collecting captive chimpanzee samples, we thank the staff at Southwick's Zoo, Mendon, MA. For assistance in carrying out experiments, we thank Cary Allen-Blevins, Rachel Berg, Andy Biewener, Meg Callahan-Beckel, Brian Hare, Kathleen Pritchett-Corning, Pedro Ramirez, and Emily Venable. For helpful comments on the manuscript, we thank Daniel Lieberman, Richard Wrangham, and members of the Carmody lab.

## Additional information

### Funding

| Funder | Grant reference number | Author |
| --- | --- | --- |
| National Institute on Aging | R01AG049395 | Melissa Emery Thompson Rachel N Carmody |
| Harvard University | Dean's Competitive Fund for Promising Scholarship | Rachel N Carmody |
| Harvard University | William F. Milton Fund | Aspen T Reese |

The funders had no role in study design, data collection and interpretation, or the decision to submit the work for publication.

### Author contributions

Aspen T Reese, Conceptualization, Formal analysis, Funding acquisition, Investigation, Visualization, Methodology, Writing - original draft, Writing - review and editing; Katia S Chadaideh, Mark Beckel, Peggy Callahan, Roberta Ryan, Investigation, Methodology, Writing - review and editing; Caroline E Diggins, Laura D Schell, Investigation, Writing - review and editing; Melissa Emery Thompson, Funding acquisition, Investigation, Writing - review and editing; Rachel N Carmody, Conceptualization, Supervision, Funding acquisition, Visualization, Methodology, Writing - original draft, Project administration, Writing - review and editing

### Author ORCIDs

Aspen T Reese  https://orcid.org/0000-0001-9004-9470
Katia S Chadaideh  https://orcid.org/0000-0003-2251-1170

Laura D Schell [ID] https://orcid.org/0000-0002-9600-924X
Rachel N Carmody [ID] https://orcid.org/0000-0001-7505-9646

### Ethics

Human subjects: Human samples were self-collected by healthy study participants after providing written informed consent under the approval of the Harvard University IRB (protocol 17-1016).

Animal experimentation: All experiments were conducted in accordance with the National Institutes of Health Guide for the Care and Use of Laboratory Animals. All mouse experiments were performed with protocols approved by the Harvard University Institutional Animal Care & Use Committee (protocol 17-11-315). All canid experimentation was approved by the WSC IACUC (protocol HAR-001). Wild chimpanzee fecal samples were collected by field assistants under the approval of the UNM IACUC (protocol 18-200739-MC) and with permission of the Uganda Wildlife Authority and Uganda National Council for Science and Technology. Fecal samples from other non-human mammals were collected from the ground following natural production; this approach precluded the need for institutional approval and was non-invasive for the animals.

### Decision letter and Author response

Decision letter https://doi.org/10.7554/eLife.60197.sa1
Author response https://doi.org/10.7554/eLife.60197.sa2

## Additional files

### Supplementary files

• Supplementary file 1. Beta-diversity and nonmetric multidimensional scaling (NMDS) shift analyses were generally robust to the distance metric used and to subsetting the dataset.

• Supplementary file 2. Expanded sampling metadata.

• Supplementary file 3. Nutritional information for experimental diets.

• Transparent reporting form

### Data availability

All sequencing data included in this study are available in the European Nucleotide Archive under accession number PRJEB36262.

The following dataset was generated:

| Author(s) | Year | Dataset title | Dataset URL | Database and Identifier |
|---|---|---|---|---|
| Reese AT, Carmody RN | 2020 | Effects of domestication on the gut microbiota parallel those of human industrialization | https://www.ebi.ac.uk/ena/browser/view/PRJEB36262 | European Nucleotide Archive, PRJEB36262 |

The following previously published dataset was used:

| Author(s) | Year | Dataset title | Dataset URL | Database and Identifier |
|---|---|---|---|---|
| Jha AR, Davenport ER, Gautam Y, Bhandari D, Tandukar S, Ng KM, Fragiadakis GK, Holmes S, Gautam GP, Leach J, Sherchand JB, Bustamante CD, Sonnenburg JL | 2018 | Gut microbiome transition across a lifestyle gradient in Himalaya | https://www.ebi.ac.uk/ena/browser/view/PRJEB29137 | European Nucleotide Archive, PRJEB29137 |

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
