## [Decision Letter]

**Acceptance summary:**

This work examines the role of domestication and industrialization on the microbiome by looking at changes in the gut microbiota of humans and wild and domesticated mammals. Despite being fundamentally different processes, the authors conclude that domestication and industrialization have impacted the gut microbiota in related ways.

**Decision letter after peer review:**

[Editors’ note: the authors submitted for reconsideration following the decision after peer review. What follows is the decision letter after the first round of review.]

Thank you for submitting your work entitled "Effects of domestication on the gut microbiota parallel those of human industrialization" for consideration by *eLife*. Your article has been reviewed by a Senior Editor, a Reviewing Editor, and three reviewers. The reviewers have opted to remain anonymous.

Our decision has been reached after consultation between the reviewers. Based on these discussions and the individual reviews below, we regret to inform you that we cannot pursue the publication of your manuscript in its present form. However, we consider that the work is very interesting and therefore a new version that thoroughly addresses the concerns raised would likely be reviewed again. Although it would be treated as a new submission, we would aim to retain an overlapping set of reviewers.

The work provides exciting and valuable information on the possible effects of domestication and industrialization on the gut microbiome. However, there were several methodological issues raised, such as host genotype determination, control for genetic distance and, in particular, concerns regarding data analyses (diversity metrics, OTU picking, error rates, Permanova, and FDR correction, to name some) that can take a considerable amount of time to perform. There were also misgivings regarding the validity of some of the conclusions based on the data presented. These include group comparisons that do not necessarily agree with the idea that domestication and industrialization similarly impact the gut microbiota, and the effect of host genotype or genetic distance on the microbiota. Please also take into account the comments about differences with respect to previous publications regarding the claim that domestic animals may be useful as models.

Reviewer #1:

This study examines effects of domestication on the gut microbiome of wild animals to the effect of industrialization on the gut microbiota of humans. They report consistent shifts in composition of gut microbiota in domestic animals and in humans from industrialized (but not from traditional) societies. They also perform cross-feeding experiments of wild and domesticated animals (lab mice/wild mice; dogs/wolves) and report that apart from genetics, diet plays a dominant role in shaping (loss of diversity) of the domestic gut microbiota.

I have the following comments:

1) Introduction: "genetic changes under domestication" How did the authors control for differences in genetic distance among the individual domesticated/wild animal pairs? Are the shifts in composition of microbiota during domestication and industrialization still consistent if controlled for genetic distance?

2) Introduction: "Finally, the convergent nature of many ecological shifts experienced by domesticated animals and industrialized human populations suggests that domestic animals may provide a uniquely useful model for studying the microbially-mediated health impacts of rapid environmental change." and Discussion "their translational potential as models for studying the gut microbiota of industrialized populations may be greater than is currently appreciated." This statement is not clear – please clarify in the context of two publications (Nature. 2016;532(7600):512-6 and Science, 708 2019;365(6452):eaaw4361) that appear to state the opposite.

3) The authors describe that diet plays a major role in changing the microbiota of wild animals to those of domestic ones. Diet is a great source of viruses. To which extend is the introduction or loss of viruses (in particular phages) responsible for the shift in gut microbiota?

4) The authors state repeatedly that wild animals more diverse microbiota. Are there uniform changes in taxa? Are some taxa lost, and if so, is this observed in several wild / domestic pairs?

Reviewer #2:

This is an exciting paper with important implications for how diet and environment interact to shape the composition and diversity of the gut microbiome. The results are interesting – particularly the results of the robustly designed diet challenge experiments. Concluding with the host-microbe-environment mismatch puzzle is thought-provoking. I am slightly concerned about how the framing of the study is phrased. I additionally have some questions/suggestions/concerns regarding the methods.

1) The results of this study are super interesting, but the authors need to be sure to make it very clear throughout that they are examining how environmental and dietary shifts associated with domestication may parallel environmental and dietary changes in some human populations (not that some human populations are domesticated and some are wild). The authors are mindful to make this clear most of the time, but it would be good to make it explicit all of the time.

In addition, I would ask that the authors carefully consider how human populations are described – traditional is not the best term, unless it is how those populations self-identify. There are real and very important ecological differences that distinguish the human populations that were sampled. Using language that somehow indicates what those differences are might be more impactful than using industrialized vs. traditional. Or, at the very least, clearly defining those terms early on in the article is necessary. Industrialized vs. non-industrialized or traditional can be read as placing as elevating either group or could be read as saying the populations are "advanced" and "not advanced" (particularly important as this paper will likely generate some media attention).

2) I am wondering if it is better to categorize the genotype/diet experiment as a provenance/diet experiment or something similar. As the authors did not actually look at host genetics in the wild-caught mice, they don't know how genetically distinct they are and there is certainly variation in genetic distance from the lab mice within the group of wild-caught mice.

3) Genetic changes kind of come up unexpectedly and without context the Introduction, which I found unclear. It may be better to focus this paragraph solely on ecological/environmental shifts? I was also a little confused if the authors were indicating the known genetic changes caused by domestication would change something about host physiology that would impact the gut microbiome somehow, or if the effect of divergence in host genetics would cause a simultaneous divergence in gut microbiome composition, or both.

4) Gomez et al., 2019 and Amato et al., 2019 both found that the human gut microbiota is actually closer to that of baboons than chimpanzees. I don't think the authors necessarily need to add baboons to the analysis, but it would be relevant to acknowledge in the discussion that chimps may or may not be the best comparison for humans.

5) Introduction (and elsewhere): I don't think domestic can be used in place of domesticated – the meanings, to me at least, are distinct.

6) Methodological concerns:

- Samples collected in RNAlater are not necessarily comparable to freshly frozen – please note in the methods which species were preserved with each method and describe how you accounted for this difference in preservation.

- Why was closed-reference OTU-picking chosen? Open-reference OTU-picking is the recommended method, unless one is comparing amplicons from different regions of the 16S rRNA gene. I would suggest that analyzing the data using one of the ASV strategies (DADA2 or Deblur) is recommended, but also do not want to force the authors to reanalyze their entire dataset (and the newer ASV methods become less useful when including 454 data).

- Yatsunenko et al., 2012 used 454 sequencing – I am curious how the authors corrected for the differences in sequencing-related error rates between 454 and HiSeq? And why they did not choose to use available human datasets sequenced in a manner comparable to the newly produced dataset in this paper?

- Using the adonis2 function in vegan would allow the authors to use marginal sums of squares in the PERMANOVA analysis – this might allow them to better tease apart which factors are accounting for what proportion of the variance in the dataset.

- A Bonferroni correction is quite conservative for microbiome datasets – FDR correction could be used instead.

- I would like to see an explanation for the choice of method to measure the magnitude of change in β-diversity, as it is one I haven't seen before and measuring change along an axis that does not have an easily interpretable meaning might not be the most informative. Alternatively, comparing pairwise unweighted and weighted UniFrac between domesticated/wild and baseline/treatment and/or performing a Procrustes analysis may be preferred.

Reviewer #3:

Reese et al., compare the microbiota of domestic animals and their closest wild counterparts, including a comparison of humans and chimpanzee microbiotas. They report similar changes to the microbiota from domestication and industrialization. Overall, the data presented is fairly noisy and many of the conclusions seem overstated given slight differences between groups. Even if we set aside the issues with the data, which are not trivial, it is unclear how important the conclusions are. For example, the last sentence of the Abstract:

"We conclude that domestication and industrialization have similarly impacted the gut microbiota, emphasizing the utility of domestic animal models and diets for understanding host-microbial interactions in rapidly changing environments, and the importance of studying non-industrialized human populations for understanding aspects of human health dependent on host-microbial co-evolution."

Not so easy to unravel the point(s) the authors are trying to make. The last passage is already very clear to the field, non-industrialised populations are important to study. The first part suggests that domestic animals and diets are useful in understanding the microbiota in changing environments. It is not clear exactly what this statement is trying to convey and it requires some clarification.

In the Abstract the authors state that "domestication and industrialization have similarly impacted the gut microbiota". A major concern is the data presented in Figure 5B for two reasons. First, the difference between two industrialized human populations appears to be larger than that observed between domestic and wild animals. Second, the shift to the left from industrialized humans to traditional humans is larger than from industrialized humans to chimpanzees. Not only is this problematic from the standpoint of implications about the "wildness" of traditional populations, but also difficult to interpret given the greater similarity in genetics, physiology, lifestyle, and diet between human populations than chimpanzees and humans.

The authors report greater between species variability in wild gut communities than domesticated. However, it does not look like they did this comparison for the human and chimpanzee data. Given published data showing that the between individual variability in the microbiota of industrial individual is larger than that of traditional population microbiota, it would be interesting to see how these data compare to that of chimpanzees given that this is not the result you would expect given the data from the other animal pairs.

It is not clear how α diversity was calculated. Was the data rarefied and if so to how many reads and were the samples sequenced sufficiently deep to ensure an accurate measurement of diversity.

Subsection “Diet vs. genotype effects on domestic gut microbial composition in mice”. "Domestication has profound effects on both ecology and host genotype." Do the authors mean "has had", ie, there is evidence that animals, when domesticated, show genotypic changes, eg, new traits are selected for. Domestication over short time periods may have little effect on genotype.

Subsection “Diet vs. genotype effects on domestic gut microbial composition in mice”. "we found that host genotype explained the largest amount of variation" It is unclear what data the authors are examining to reach this conclusion. The species appears to be *Mus musculus* for these analyses. Are the authors performing a host genotype (eg, SNP) analysis? Please clarify how differences in host genotype are being determined.

Figure 2.

- It is very difficult to draw conclusions from Figure 2B. Suggest that the authors show centroids or find a better way to represent the data. Some of the colors are too similar as well, so difficult to differentiate. Why are DomG/DomD points moving on the PCA plot? Same with WildG/WildD? Perhaps this data could reveal drift of the microbiome composition in the absence of intervention, which may inform whether their diet shift in the other groups is meaningful.

Figure 4 has many of the same issues described for Figure 2. It's very difficult to interpret these panels with so many points going in different direction and minimal color differences between some of the points.

[Editors’ note: further revisions were suggested prior to acceptance, as described below.]

Thank you for submitting your article "Effects of domestication on the gut microbiota parallel those of human industrialization" for consideration by *eLife*. Your article has been reviewed by Detlef Weigel as the Senior Editor, a Reviewing Editor, and three reviewers. The reviewers have opted to remain anonymous.

The reviewers have discussed the reviews with one another and the Reviewing Editor has drafted this decision to help you prepare a revised submission.

Summary:

The manuscript by Reese et al., explores the effects of mammalian domestication and human industrialization on the gut microbiota and has important implications on how diet and environment interact to shape the composition and diversity of the gut microbiome. They characterize the microbiome via 16S rRNA gene sequence analysis in various mammalian species and show that the microbiome shifts with domestication. Using cross-feeding experiments in mice, wolves and dogs they are able to demonstrate that diet and microbial inoculation can reverse the effects of domestication. Finally, they compare chimpanzees and both industrialized and non-industrialized human populations and show that shifts in microbiome composition are larger when chimpanzees are compared with industrialized populations. Overall the work presents clever experiments aimed at characterizing the effects of domestication on the gut microbiota and comparing these effects with those of human industrialization.

Essential revisions:

1) Reviewers were concerned with the comparisons between domestication and industrialization and the subsequent conclusions. This aspect of the work needs to be improved for clarity and the claims toned down as they are not fully supported by the data presented.

a) The authors should note that domestication, which has taken a long time, and industrialization, a fairly recent change to our ecology, are different processes. Therefore, the direct comparisons in the manuscript do not seem entirely appropriate and should be more carefully addressed. In particular, the data does not provide strong evidence to support the claim that animal domestication and human industrialization result in similar effects on their hosts microbiome, even though this conclusion may be correct, since it makes sense given that many ecological processes are probably affected in similar ways. This conclusion should therefore be toned down to agree with their data.

b) Is there a way to incorporate data from populations that use subsistence strategies involving domestication, but are not Industrialized (the other populations in Jha et al., even)? It could be expected that the agricultural or pastoral but non-Industrialized countries would be somewhat intermediate in their microbiome composition, as they experience the factors of domestication without some of the extreme ecological consequences of Industrialization (antibiotics, highly processed foods, etc.). Is this the case?

c) A more nuanced discussion should occur at some point in the manuscript on the choice and caveats of using highly Industrialized populations in this comparison given that the process being compared is domestication and not industrialization.

2) The revised manuscript has improved but still lacks clarity in many places and uses language that is vague and often misleading, making it difficult to understand what the authors are trying to say. The entire text should therefore be checked and improved to make the language more precise.

a) In the Abstract, for example, it is not clear what shifts the authors refer to, what is meant by microbiomes to be impacted “'similarly”, and what “parallel ecological changes” are. It can be argued that the ecological changes are quite different in industrialized humans and domesticated animals (housing, hygiene, diet, etc.). However, the ecological processes that impacted their microbiomes, and the compositional alterations, might have been similar.

b) This vagueness is also found through the entire manuscript. What are ecological parallels (Introduction)? What is a "suite of shared ecological changes" (Introduction)? Which “evolutionary forces” were studied? What do the authors mean by "individual shifts"? (figure legend of Figure 1C). Compositional shifts in an individual? Was that even assessed?

c) The term “shifts” is used inappropriately throughout the manuscript. For example, what are "shifts between industrialized humans and wild chimpanzees" (Figure 5 legend)? The microbiome does not really shift from a human to a chimpanzee. Do the authors refer to differences between microbiomes in different hosts?

3) The authors should be careful with the way they present their results to avoid biased interpretation and make claims that are clearly supported by their results.

a) It sometimes seems as if the authors have interpreted the findings to fit a preconceived idea of the findings. For example, the authors conclude a "consistent effect of domestication status" (Results), but the samples cluster by host, which has the highest effect sizes. The conclusion is then mainly based on a statistical analysis that showed domesticated samples to be "further right" on an NMDs axis. This is not very convincing, and not very clear in Figure 1C either.

b) Another claim is that in Figure 2, differences between domesticated and wild mice can be overcome by a diet switch, but looking at Figure 2—figure supplement 2, that is simply not the case. It is difficult to see how the data in Figure 5 provides strong evidence that the effects of domestication and industrialization are similar.

4) More clarification is needed for wild and domestic microbiome results and subsequent conclusions

a) The results presented (Results and Figure 1) do not seem to support the conclusion that domestication is shifting all species to the right along NMDS1. The magnitude and direction of shift seems to differ based on host species. While the general trend of all species lumped together is to the right, sheep and pigs don't seem to follow the pattern (and some others don't seem to have a strong shift to the right). What are the effect sizes for the Mann-Whitney U tests here? Also, looking at Figure 1—figure supplement 2A, only the companion species are denoted as having a p<0.05, which seems at odds with the statement in the Results.

This species-dependent direction and strength of shift is not entirely unexpected based on previous work. Shifting host ecology (diet or captivity) has previously been shown to differentially effect host species: Amato et al., 2015 and McKenzie et al., 2017.

The inconsistency in the direction of the shift might not actually negate the broader point, that domestication at times has effects on the gut microbiome that are very similar to the shift we see between industrialized and non-industrialized humans. In fact, it might be instructive to point out what specific species might be good models for the shift we see in humans – what are the specific ecological shifts with domestication in those species and how does that mirror the ecological shift with industrialization in humans?

b) What does it look like when you put the results of the mouse and canid experiments in the same ordination space with your wild/domestic and chimp/human pairs? Is the shift in the expected direction? When looking at the results of the mouse experiment and the canid experiments on their own, we see a shift to the left with experimental domestication (ie, for the Wildh/Domd treatments), but this might be a function of the ordination space?

c) Were any of the animals, either wild or domestic, from the same family, field, pen, etc.? Cohousing results in convergent microbiome profiles across a number of species due to horizontal microbial exchange. If conspecifics were collected from the same living situation or were related, one might expect higher microbiome sharing on those grounds alone. This potential confounder could explain the high similarity between the conspecifics. These details should be added to the Materials and methods. If this is an issue, it should be corrected for in statistical comparisons (if possible).

5) Technical concerns and data presentation

a) Figure 2 and Figure 4 are a difficult to interpret, because the lines used to indicate moving points are obscuring the points themselves in some cases. Would ellipses around the treatment groups in the NMDS plots be more informative than the moving points?

b) For the adonis2 function, to get the marginal sums of squares you need to include “by = "margin"” in the function call. Using adonis2 without specifying “by” is equivalent to using the older adonis function. This should be relatively quick to rerun and will make the effect of host vs. ecology vs. diet easier to parse.

c) The OTU picking strategy can introduce biases when sampling microbes that are better represented in the reference taxonomy since more of the sequences will be classified in one sample versus another. Even though the authors seem to have chosen the best option for this dataset, there very well could be differences given that comparisons are explicitly between Industrialized vs Non-Industrialized populations (there tends to be lower read mapping to closed ref OTUs in non-industrial populations) as well as human-associated vs. wild animals (it would be expected that lab animals and livestock microbiomes have been better characterized back when that GreenGenes taxonomy was created).

Can a Supplementary file be added that lists the proportion of reads classified per sample? Are there differences in the number of reads that classify between the major comparisons in this paper (Industrialized vs. Non-Industrialized, Wild vs. Domestic, etc.)? If there are, then reprocessing of these reads either with an open OTU calling method or ASV method should be implemented.

d) How does microbial load/density vary based on gut passage rates, and could this be influencing your results?

---

## [Author Response]

Our decision has been reached after consultation between the reviewers. Based on these discussions and the individual reviews below, we regret to inform you that we cannot pursue the publication of your manuscript in its present form. However, we consider that the work is very interesting and therefore a new version that thoroughly addresses the concerns raised would likely be reviewed again. Although it would be treated as a new submission, we would aim to retain an overlapping set of reviewers.The work provides exciting and valuable information on the possible effects of domestication and industrialization on the gut microbiome. However, there were several methodological issues raised, such as host genotype determination, control for genetic distance and, in particular, concerns regarding data analyses (diversity metrics, OTU picking, error rates, Permanova, and FDR correction, to name some) that can take a considerable amount of time to perform. There were also misgivings regarding the validity of some of the conclusions based on the data presented. These include group comparisons that do not necessarily agree with the idea that domestication and industrialization similarly impact the gut microbiota, and the effect of host genotype or genetic distance on the microbiota. Please also take into account the comments about differences with respect to previous publications regarding the claim that domestic animals may be useful as models.

The author and editorial comments were very helpful in improving our manuscript. We provide a point by point response below but have targeted our efforts towards those specified above including clarifying genotype grouping, adding new relatedness-informed analyses, and expanding/altering the methods to address the statistical and data processing concerns. We have also updated the human group analysis with a new set of publicly available data that better matches our sequencing methods, as well as new data on captive chimpanzees. Our updated results confirm or clarify our previous findings, and we think solidify our overall result that domestication and industrialization similarly influence the gut microbiota.

Reviewer #1:This study examines effects of domestication on the gut microbiome of wild animals to the effect of industrialization on the gut microbiota of humans. They report consistent shifts in composition of gut microbiota in domestic animals and in humans from industrialized (but not from traditional) societies. They also perform cross-feeding experiments of wild and domesticated animals (lab mice/wild mice; dogs/wolves) and report that apart from genetics, diet plays a dominant role in shaping (loss of diversity) of the domestic gut microbiota.I have the following comments:1) Introduction: "genetic changes under domestication" How did the authors control for differences in genetic distance among the individual domesticated/wild animal pairs? Are the shifts in composition of microbiota during domestication and industrialization still consistent if controlled for genetic distance?

In response to this comment and similar queries from reviewer 2 and reviewer 3, we have added additional analyses of genetic effects (i.e. time since divergence and time since domestication) to the cross-species comparison, and have altered our discussion of genetics in the diet experiments. Because we do not have individual genotype data for the animals included in the cross-species comparison, we cannot explicitly include a correction for relatedness to our β-diversity analyses (i.e. the PERMANOVA tests and permutation test for F). However, we have added a Mantel test result describing the relationship between β-diversity and relatedness as well as additional analyses assessing the effects of relatedness to dyad-level trends. See the Results.

“Domestication of mammalian species occurred at different times, so the evolutionary relationships between members of a host dyad, even in cases where we sampled the known progenitors, are not all equal. […] Moreover, differences in position along NMDS axis 1 remained significant even when correcting for host dyad and divergence time (P<0.001, likelihood test linear mixed effects models).”

2) Introduction: "Finally, the convergent nature of many ecological shifts experienced by domesticated animals and industrialized human populations suggests that domestic animals may provide a uniquely useful model for studying the microbially-mediated health impacts of rapid environmental change." and Discussion "their translational potential as models for studying the gut microbiota of industrialized populations may be greater than is currently appreciated." This statement is not clear – please clarify in the context of two publications (Nature. 2016;532(7600):512-6 and Science, 708 2019;365(6452):eaaw4361) that appear to state the opposite.

We appreciate the confusion these statements may have caused. To remedy and clarify, we have adjusted our discussion of animal models in both the Introduction and the Discussion. In the Introduction, we now emphasize the relevance of animal models to studies of ecological change rather than human health/biology specifically.

“Finally, the convergent nature of many ecological shifts experienced by domesticated animals and industrialized human populations suggests that domesticated animals may provide a uniquely useful model for studying the microbially-mediated health impacts of rapid environmental change resulting in mismatch between host, microbiota and/or environment, a situation thought to apply to humans in industrialized settings (32).”

In the Discussion, we discuss more extensively why lab animals may or may not be good models for humans. We present the evidence that mice are poor models for humans, including the dirty mouse references the reviewer recommends, and speculate that they may be better models for other aspects of human biology.

“Because laboratory animals demonstrate some of the largest overall differences relative to their wild counterparts, they might be expected to have high translational potential as models for studying the gut microbiota of industrialized human populations. […] Certainly, studies of non-domesticated animals will be necessary to understand natural host-microbe interactions and their evolutionary history (48, 54), as well as to determine the most appropriate models for translational research.”

3) The authors describe that diet plays a major role in changing the microbiota of wild animals to those of domestic ones. Diet is a great source of viruses. To which extend is the introduction or loss of viruses (in particular phages) responsible for the shift in gut microbiota?

This is an important and interesting point. We have added a paragraph to the Discussion outlining potential mechanisms for diet effects on the microbiome, including virus impacts.

“Our reciprocal diet experiments in mice and canids confirm that ecology plays a predominant role in shaping the domesticated gut microbiota. […] That the gut microbial impacts of change in a single ecological variable like diet were sufficiently profound to outweigh those of host taxon identity suggests that suites of ecological variables changing together, such as during domestication or industrialization, may have collectively exerted an even larger influence (36).”

4) The authors state repeatedly that wild animals more diverse microbiota. Are there uniform changes in taxa? Are some taxa lost, and if so, is this observed in several wild / domestic pairs?

We found that there are not consistent effects of domestication on α-diversity in the gut microbiota (see Figure 1—figure supplement 4D, E). On a host level, some wild progenitors have higher diversity (e.g. mice; Figure 1—figure supplement 2C), whereas some domesticated species have higher diversity than their wild progenitors (e.g. wolves; Figure 1—figure supplement 4C). We discuss some of potential mechanisms underlying this in the text. In light of this complexity, we do not necessarily see a lot of consistent differences in microbial relative abundance between wild and domestic taxa in our cross-species comparison. Some of the microbial taxon responses are shown in Figure 5—figure supplement 1 and are discussed in light of trends observed in industrialized versus non-industrialized human populations.

Reviewer #2:This is an exciting paper with important implications for how diet and environment interact to shape the composition and diversity of the gut microbiome. The results are interesting – particularly the results of the robustly designed diet challenge experiments. Concluding with the host-microbe-environment mismatch puzzle is thought-provoking. I am slightly concerned about how the framing of the study is phrased. I additionally have some questions/suggestions/concerns regarding the methods.1) The results of this study are super interesting, but the authors need to be sure to make it very clear throughout that they are examining how environmental and dietary shifts associated with domestication may parallel environmental and dietary changes in some human populations (not that some human populations are domesticated and some are wild). The authors are mindful to make this clear most of the time, but it would be good to make it explicit all of the time.In addition, I would ask that the authors carefully consider how human populations are described – traditional is not the best term, unless it is how those populations self-identify. There are real and very important ecological differences that distinguish the human populations that were sampled. Using language that somehow indicates what those differences are might be more impactful than using industrialized vs. traditional. Or, at the very least, clearly defining those terms early on in the article is necessary. Industrialized vs. non-industrialized or traditional can be read as placing as elevating either group or could be read as saying the populations are "advanced" and "not advanced" (particularly important as this paper will likely generate some media attention).

We appreciate the reviewer’s recommendation for nuance and clarity and have tried to improve treatment throughout the text. In particular we have altered our introductory presentation of different human populations (see the Introduction) and now refer to industrialized and non-industrialized throughout (rather than traditional) to keep in line with other recent work on variation in the human gut microbiota.

“Industrialized, agrarian, and foraging human populations differ along numerous ecological dimensions, including diet, physical activity, the size and nature of social networks, pathogen exposure, types and intensities of medical intervention, and altered reproductive patterns. Such changes have resulted in large shifts in the gut microbiota in industrialized populations relative to non-industrialized populations or closely related primates (1-4), including reductions in alpha-diversity and changes in composition that have been implicated in the rise of various metabolic and immunological diseases (5-7).”

2) I am wondering if it is better to categorize the genotype/diet experiment as a provenance/diet experiment or something similar. As the authors did not actually look at host genetics in the wild-caught mice, they don't know how genetically distinct they are and there is certainly variation in genetic distance from the lab mice within the group of wild-caught mice.

The reviewer is correct that we did not have individual genotype data for these individuals and thus the genotype/diet nomenclature could be confusing. As such, we have changed our presentation of these experiments to be tests of the effects of host taxon and diet. The abbreviations used in the text and figures now refer to Wild_H_/Wild_D_ etc.

3) Genetic changes kind of come up unexpectedly and without context the Introduction, which I found unclear. It may be better to focus this paragraph solely on ecological/environmental shifts? I was also a little confused if the authors were indicating the known genetic changes caused by domestication would change something about host physiology that would impact the gut microbiome somehow, or if the effect of divergence in host genetics would cause a simultaneous divergence in gut microbiome composition, or both.

In an effort to clarify this section and reduce confusion, we have edited this text, it reads:

“Many of the altered ecological features experienced by industrialized humans and domesticated animals have been independently observed to impact the gut microbiota, including diet (9, 10), physical activity (11, 12), the size and nature of social networks (13, 14), antibiotic use (15, 16), and changes in birthing and lactation practices (15, 17). These effects are often found to match or exceed the effects of genetic variation—also introduced by domestication—on gut microbiota composition (10). As such, ecological shifts under domestication might be expected to lead to microbial differentiation between domesticated animals and their wild counterparts…”

4) Gomez et al., 2019 and Amato et al., 2019 both found that the human gut microbiota is actually closer to that of baboons than chimpanzees. I don't think the authors necessarily need to add baboons to the analysis, but it would be relevant to acknowledge in the discussion that chimps may or may not be the best comparison for humans.

We are unable to add baboons to the analysis at this time because we did not have access to such samples and sequences from baboons were not included in Jha et al., the reference we used for comparative analyses. However, we do now refer specifically to these findings: (Discussion).

“While we limited our analysis to human-chimpanzee comparisons because *Pan* is our closest sister clade to *Homo*, recent work has shown that the human gut microbiota is more similar to that of baboons (50, 51). Baboons are more distantly related to humans but have been argued to be more similar in terms of diet and dietary physiology (52, 53), accentuating our finding of the importance of ecological factors in shaping the microbiota. Further work will be required to illuminate the specific combination of ecological factors driving similarities between domesticated and industrialized gut microbial signatures.”

5) Introduction (and elsewhere): I don't think domestic can be used in place of domesticated – the meanings, to me at least, are distinct.

We appreciate the reviewer’s recommendation on this point and have replaced “domestic” with “domesticated” throughout.

6) Methodological concerns:- Samples collected in RNAlater are not necessarily comparable to freshly frozen – please note in the methods which species were preserved with each method and describe how you accounted for this difference in preservation.

We have added text to the Materials and methods specifying which samples were preserved with RNAlater and (now in the case of the new human data) OMNIgene kits. We do not correct for differences in preservation in our analyses but have found that it does contribute substantially to the variation we observe. Whether analyzing all samples (including previously published human samples collected with the OMNIgene kits) or just the cross-species dataset, we find that preservative method is significant factor (PERMANOVA P<0.001) but with an R^2^=0.01. This result is outlined in the Materials and methods section describing preservation method.

- Why was closed-reference OTU-picking chosen? Open-reference OTU-picking is the recommended method, unless one is comparing amplicons from different regions of the 16S rRNA gene. I would suggest that analyzing the data using one of the ASV strategies (DADA2 or Deblur) is recommended, but also do not want to force the authors to reanalyze their entire dataset (and the newer ASV methods become less useful when including 454 data).

We are unfortunately not in a position to totally reanalyze the dataset at this point although we appreciate that there are potential benefits in using an ASV approach going forward. Here, we relied on closed-reference OTU-picking to limit batch effects between sequencing runs and preservative methods. Closed-reference OTU-picking is a common approach that is more conservative and thus unlikely to introduce spurious results when dealing with high variability datasets such as our cross-species comparison.

- Yatsunenko et al., 2012 used 454 sequencing – I am curious how the authors corrected for the differences in sequencing-related error rates between 454 and HiSeq? And why they did not choose to use available human datasets sequenced in a manner comparable to the newly produced dataset in this paper?

The reviewer is correct that the difference in sequencing platforms between the published data and our own is significant. With that in mind we have replaced our usage of data from Yatsunenko et al., with more recent data from Jha et al., 2018. This new data was produced with the same primers as we used for our data collection and was also sequenced on the Ilumina platform. To maximize the comparability of their data and ours, we reprocessed the raw sequences from Jha et al. using our bioinformatics pipeline. There are limitations to this dataset as well, of course—in particular some of the samples were collected with OMNIgene kits and stored in preservative before DNA extraction—but we include new text describing these.

- Using the adonis2 function in vegan would allow the authors to use marginal sums of squares in the PERMANOVA analysis – this might allow them to better tease apart which factors are accounting for what proportion of the variance in the dataset.

Thank you for the recommendation. We have replaced all PERMANOVA results with output from the adonis2 function rather than the adonis function.

- A Bonferroni correction is quite conservative for microbiome datasets – FDR correction could be used instead.

The reviewer is right that Bonferroni is quite conservative. However, in this case, FDR and Bonferroni corrections did not produce substantially different outcomes so we have erred on the side of conservatism by retaining the Bonferroni correction throughout.

- I would like to see an explanation for the choice of method to measure the magnitude of change in β-diversity, as it is one I haven't seen before and measuring change along an axis that does not have an easily interpretable meaning might not be the most informative. Alternatively, comparing pairwise unweighted and weighted UniFrac between domesticated/wild and baseline/treatment and/or performing a Procrustes analysis may be preferred.

We have added text to the methods and results describing the utility of this metric for analyzing change in microbial community composition. It is comparable to an analysis employed by Ang et al., 2020 highlighted in Figure 1C. We have retained this analysis, despite its relative novelty, because it allows us to consider the direction of change in a way that comparing pairwise dissimilarity measurements does not. However, additional analyses on such measurements have also been added for the cross-species comparison (i.e., Figure 1D).

Reviewer #3:Reese et al., compare the microbiota of domestic animals and their closest wild counterparts, including a comparison of humans and chimpanzee microbiotas. They report similar changes to the microbiota from domestication and industrialization. Overall, the data presented is fairly noisy and many of the conclusions seem overstated given slight differences between groups. Even if we set aside the issues with the data, which are not trivial, it is unclear how important the conclusions are. For example, the last sentence of the Abstract:"We conclude that domestication and industrialization have similarly impacted the gut microbiota, emphasizing the utility of domestic animal models and diets for understanding host-microbial interactions in rapidly changing environments, and the importance of studying non-industrialized human populations for understanding aspects of human health dependent on host-microbial co-evolution."Not so easy to unravel the point(s) the authors are trying to make. The last passage is already very clear to the field, non-industrialised populations are important to study. The first part suggests that domestic animals and diets are useful in understanding the microbiota in changing environments. It is not clear exactly what this statement is trying to convey and it requires some clarification.

Throughout the revised manuscript, we have attempted to be more concrete about what can and cannot be concluded from our data. In addition, we have edited the final sentence of the Abstract to improve clarity, it reads:

“Our findings emphasize the utility of domesticated animal models for understanding hostmicrobial interactions in rapidly changing environments, while highlighting the limitations of animal models and the importance of studying non-industrialized human populations for understanding aspects of biology dependent on host-microbial co-evolution.”

In the Abstract the authors state that "domestication and industrialization have similarly impacted the gut microbiota". A major concern is the data presented in Figure 5B for two reasons. First, the difference between two industrialized human populations appears to be larger than that observed between domestic and wild animals. Second, the shift to the left from industrialized humans to traditional humans is larger than from industrialized humans to chimpanzees. Not only is this problematic from the standpoint of implications about the "wildness" of traditional populations, but also difficult to interpret given the greater similarity in genetics, physiology, lifestyle, and diet between human populations than chimpanzees and humans.

We sincerely thank the reviewer for their sensitivity to potential (though unintended) interpretations. In response to this reviewer’s concerns and those of reviewer 2, we have updated our analyses on different human populations. Using a more recent dataset, with samples prepared in a manner more in line with our own methods, we have found generally the same results, with trends between industrialized populations and chimps paralleling those between domesticated and wild animals (see updated Figure 5). We now find that the non-industrialized populations (now a group of foragers from Nepal and a group of hunter-gatherers from Tanzania) do not differ in the opposite direction but instead are just in line with the chimpanzees. Furthermore, we do not observe a large difference between the two industrialized human populations. This updated analyses should address the concerns the reviewer outlines above. In addition, to help guard against readers inferring any inappropriate distinctions between industrialized and non-industrialized populations, we have made revisions throughout the text to emphasize that these populations are equally human and that their differences are primarily rooted in different ecologies.

The authors report greater between species variability in wild gut communities than domesticated. However, it does not look like they did this comparison for the human and chimpanzee data. Given published data showing that the between individual variability in the microbiota of industrial individual is larger than that of traditional population microbiota, it would be interesting to see how these data compare to that of chimpanzees given that this is not the result you would expect given the data from the other animal pairs.

We appreciate this recommendation and have added a variability analysis for the human/chimpanzee comparison using the new data that we produced here (7 wild chimpanzees and 7 US human adults). For these samples at least, we do not find a significant difference in variability. The result is described in the text:

“We also found a marginal difference in between-conspecific variability in the gut microbiota of humans and chimpanzees (P=0.092, F=3.0987; permutation test for F). “

It is not clear how α diversity was calculated. Was the data rarefied and if so to how many reads and were the samples sequenced sufficiently deep to ensure an accurate measurement of diversity.

We apologize for any confusion around our methods. We have added greater detail on how α-diversity was calculated in the Materials and methods and now include rarefaction read depths for each dataset:

“Alpha-diversity (Shannon index, OTU richness) were calculated for rarefied OTU tables (rarefaction limit of 17,500 for cross-species dataset; 27,000 for wild mouse study; 15,500 for the mouse colonization study; 7,500 for canid experiment).”

Subsection “Diet vs. genotype effects on domestic gut microbial composition in mice”. "Domestication has profound effects on both ecology and host genotype." Do the authors mean "has had", ie, there is evidence that animals, when domesticated, show genotypic changes, eg, new traits are selected for. Domestication over short time periods may have little effect on genotype.

This is a fair point. We have edited the sentence to read: “Domestication has had profound effects on both ecology and host genotype.”

Subsection “Diet vs. genotype effects on domestic gut microbial composition in mice”. "we found that host genotype explained the largest amount of variation" It is unclear what data the authors are examining to reach this conclusion. The species appears to be *Mus musculus* for these analyses. Are the authors performing a host genotype (eg, SNP) analysis? Please clarify how differences in host genotype are being determined.

The reviewer is correct that we did not collect individual host genotyping for the mice or canids. In response to this comment and one from reviewer 2 (R2C2), we have altered our presentation of these experiments to be comparisons of the effects of host-taxon and diet rather than genotype and effect.

Figure 2.- It is very difficult to draw conclusions from Figure 2B. Suggest that the authors show centroids or find a better way to represent the data. Some of the colors are too similar as well, so difficult to differentiate. Why are DomG/DomD points moving on the PCA plot? Same with WildG/WildD? Perhaps this data could reveal drift of the microbiome composition in the absence of intervention, which may inform whether their diet shift in the other groups is meaningful.Figure 4 has many of the same issues described for Figure 2. It's very difficult to interpret these panels with so many points going in different direction and minimal color differences between some of the points.

We thank the reviewer for highlighting the issues with clarity. We have now reformatted the figures to better highlight the patterns of interest, e.g. to more clearly demonstrate the baseline and endpoint points. It is typical for control animals to show some drift over the course of an experiment due both to natural biological turnover and sequencing noise. This is likely what we are observing in the Dom_H_/Dom_D_ and Wild_H_/Wild_D_ animals. However, we do not find that this variation is directional (e.g. control individuals are not moving in a consistent direction and thus do not have a shift significantly different from 0), unlike the patterns observed in the reciprocal diet treatment groups. We have tried to clarify this finding in the results and have redrawn the plots to more clearly demonstrate the baseline and endpoint points for each individual.

[Editors’ note: further revisions were suggested prior to acceptance, as described below.]

Essential revisions:1) Reviewers were concerned with the comparisons between domestication and industrialization and the subsequent conclusions. This aspect of the work needs to be improved for clarity and the claims toned down as they are not fully supported by the data presented.a) The authors should note that domestication, which has taken a long time, and industrialization, a fairly recent change to our ecology, are different processes. Therefore, the direct comparisons in the manuscript do not seem entirely appropriate and should be more carefully addressed. In particular, the data does not provide strong evidence to support the claim that animal domestication and human industrialization result in similar effects on their hosts microbiome, even though this conclusion may be correct, since it makes sense given that many ecological processes are probably affected in similar ways. This conclusion should therefore be toned down to agree with their data.

We now explicitly state in the Abstract and first paragraph of the Introduction that domestication and industrialization are fundamentally distinct processes.

“Although industrialization and domestication are fundamentally different processes, the ecological parallels between human industrialization and animal domestication suggest that the gut microbiota of diverse domesticated animals may differ in consistent ways from those of their wild progenitors, and further, that their differences may resemble those observed between industrialized and non-industrialized human populations.”

In this manuscript, our dual consideration of domestication and industrialization is solely to test for analogous effects. We have now endeavored to clarify this rationale throughout the manuscript. For example, the title of the section on the comparisons is now “Analogous pressures in the human gut microbiota”. Some other modifications of the text that emphasize the distinction between the processes of domestication and industrialization include:

“While we focus primarily on the impacts of domestication on the mammalian gut microbiota, we include analyses of industrialized and non-industrialized human populations because much is known about the effects of industrialization on the gut microbiota and as such it can serve as a benchmark ecological process for domestication. In addition, to explore the extent to which deeper evolutionary history affects these patterns, we also compare humans to chimpanzees (*Pan troglodytes*), one of our two closest living relatives and arguably the better referential model for the last common ancestor between *Pan* and *Homo* (23).”

“Finally, the convergent nature of many ecological shifts experienced by domesticated animals and industrialized human populations suggests that domesticated animals may provide a uniquely useful model for studying the microbially-mediated health impacts of rapid environmental change that results in mismatch between host, microbiota and/or environment, a situation thought to apply to humans in industrialized settings (36). Understanding what shapes the domesticated microbiota may therefore identify routes to improve experimental models, animal condition, and human health.”

“We next explored the extent to which humans harbor gut microbial signatures analogous to those of domestication. […] We first compared samples that we collected from industrialized humans and wild chimpanzees, finding that the gut microbial communities of these humans and chimpanzees exhibited differences that paralleled those observed between domesticated animals and their wild counterparts when compared in the same ordination space (P<0.001, Mann-Whitney U test; Figure 5A, 5B).”

“We observed some correspondence between the gut microbial signatures of animal domestication and human industrialization that is most likely attributable to convergent ecological changes. The observation that gut microbial divergence among *Pan* and *Homo* primarily affects industrialized populations specifically implicates recent ecological changes as opposed to either ecological changes with deeper roots in human evolution or host evolutionary changes.”

b) Is there a way to incorporate data from populations that use subsistence strategies involving domestication, but are not Industrialized (the other populations in Jha et al., even)? It could be expected that the agricultural or pastoral but non-Industrialized countries would be somewhat intermediate in their microbiome composition, as they experience the factors of domestication without some of the extreme ecological consequences of Industrialization (antibiotics, highly processed foods, etc.). Is this the case?

We appreciate this recommendation to test for industrialization effects in a more nuanced way. Following the reviewers’ suggestion, we now include a subsample of individuals from all of the Jha et al. Nepalese populations in addition to their Hadza and US populations. Notably, we added samples from two populations transitioning to subsistence farming and a population that transitioned to subsistence farming within the past two centuries. We concur with the reviewers that these additional populations could in theory experience aspects of ecology more similar to the industrialized/domesticated environment, such as exposure to domesticated animals and diets higher in grains, and thus could be intermediate in their microbiota composition. However, we found that all three populations clustered with the traditional populations previously included (Hadza and Chepang from Nepal) and that they did not display the ordination shifts associated with domestication. The text and figures (Figure 5 and Figure 5—figure supplement 1) have been updated to reflect this finding and we further emphasize that extreme ecological shifts associated with industrialization and domestication are likely the drivers of the shared pattern.

“We observed some correspondence between the gut microbial signatures of animal domestication and human industrialization that is most likely attributable to convergent ecological changes. […] These factors would be absent even in populations currently undergoing the transition from subsistence to industrialized lifestyles (39), but may overlap with changes experienced by domesticated animals in their diets, habitats, and social milieu.”

c) A more nuanced discussion should occur at some point in the manuscript on the choice and caveats of using highly Industrialized populations in this comparison given that the process being compared is domestication and not industrialization.

Throughout the manuscript, we have made changes to emphasize that our primary focus is on domestication and not industrialization, including the following prominent statement in the Introduction:

“Here, we assess the effects of domestication on the mammalian gut microbiota, perform controlled dietary experiments that attempt to distinguish between the relative roles of ecology and genetics in driving these patterns, and compare the effects of domestication to those of human industrialization. While we focus primarily on the impacts of domestication on the mammalian gut microbiota, we include analyses of industrialized and non-industrialized human populations because much is known about the effects of industrialization on the gut microbiota and as such it can serve as a benchmark ecological process for domestication.”

We have also added text to the Materials and methods describing our choice of populations and our intent in analyzing industrialization as an ecological analog to domestication:

“These populations represent extremes of industrialized and nonindustrialized human lifestyles with the variation among the non-industrialized groups not covering the full breadth of intermediate lifestyles (e.g. modern agricultural or recent urban transplants). We believe these extremes enable us to test the how human gut microbial communities respond to major ecological change of a magnitude that could be argued to approximate that experienced by gut microbial communities of animals undergoing domestication.”

2) The revised manuscript has improved but still lacks clarity in many places and uses language that is vague and often misleading, making it difficult to understand what the authors are trying to say. The entire text should therefore be checked and improved to make the language more precise.a) In the Abstract, for example, it is not clear what shifts the authors refer to, what is meant by microbiomes to be impacted “'similarly”, and what “parallel ecological changes” are. It can be argued that the ecological changes are quite different in industrialized humans and domesticated animals (housing, hygiene, diet, etc.). However, the ecological processes that impacted their microbiomes, and the compositional alterations, might have been similar.

We have attempted to clarify the language in the Abstract, now specifying some of the types of ecological changes considered and noting that there were analogous, not identical, changes between domestication and industrialization. “Domesticated animals experienced profound changes in diet, environment, and social interactions that likely shaped their gut microbiota and were potentially analogous to ecological changes experienced by humans during industrialization… Although fundamentally different processes, we conclude that domestication and industrialization have impacted the gut microbiota in related ways, likely through shared ecological change.”

b) This vagueness is also found through the entire manuscript. What are ecological parallels (Introduction)? What is a "suite of shared ecological changes" (Introduction)? Which “evolutionary forces” were studied? What do the authors mean by "individual shifts"? (figure legend of Figure 1C). Compositional shifts in an individual? Was that even assessed?

In addition to altering the abstract, we have made edits throughout the manuscript in an attempt to minimize vagueness and increase reader comprehension. For instance, we have removed the term “ecological parallels” referenced above, and we now instead specify the types of changes expected to be common to both industrialized and domesticated populations.

“Industrialized, agrarian, and foraging human populations differ along numerous ecological dimensions, including diet, physical activity, the size and nature of social networks, pathogen exposure, types and intensities of medical intervention, and reproductive patterns. […] Although industrialization and domestication are fundamentally different processes, the ecological parallels between human industrialization and animal domestication suggest that the gut microbiota of diverse domesticated animals may differ in consistent ways from those of their wild progenitors, and further, that their differences may resemble those observed between industrialized and non-industrialized human populations.”

We have removed “suite of shared ecological changes” referenced above. The sentence now reads:

“Apart from the pressures of ecological change that domestic animals experience in human environments, animal domestication has also entailed strong artificial selection for phenotypes desirable to humans, such as rapid growth and docility in agricultural animals, reliable reproduction and stress resistance in laboratory animals, and unique physical and/or behavioral attributes in companion animals.”

We have replaced “explore the ecological and evolutionary forces” referenced above. The sentence now reads:

“Here, we assess the effects of domestication on the mammalian gut microbiota, perform controlled dietary experiments that attempt to distinguish between the relative roles of ecology and genetics in driving these patterns, and compare the effects of domestication to those of human industrialization. While we focus primarily on the impacts of domestication on the mammalian gut microbiota, we include analyses of industrialized and non-industrialized human populations because much is known about the effects of industrialization on the gut microbiota and as such it can serve as a benchmark ecological process for domestication.”

We have also sought to more clearly define the ordination shift variable (both in the text and in the figure legends) and articulate why it was included. The point of first introduction of this variable in the text and the relevant section of the Figure 1 legend are copied below.

“To determine whether there was a consistent change in microbial composition with domestication, we calculated the difference between an individual’s ordination coordinates and the average ordination coordinates of its host dyad along the first nonmetric multidimensional scaling (NMDS) axis. Quantifying this ordination shift allowed us to consider overall changes in composition while correcting for host dyad and retaining information on the directionality of changes.”

Figure 1 legend: “Fig. 1. The mammalian gut microbiota carries a global signature of domestication… (C) Distance to dyad (color) mean along Bray-Curtis ordination NMDS axis 1 differs by domestication status (P=0.006, Mann-Whitney U test).”

c) The term “shifts” is used inappropriately throughout the manuscript. For example, what are "shifts between industrialized humans and wild chimpanzees" (Figure 5 legend)? The microbiome does not really shift from a human to a chimpanzee. Do the authors refer to differences between microbiomes in different hosts?

Throughout the manuscript, we have replaced the term “shift” with the phrase “ordination shift” to clarify that we are not considering ties in real world, just in analytical space. The term shift is frequently used to describe such ordination-based findings (e.g. see McKenzie et al., 2017 and Ang et al., 2020), but we believe that stipulating “ordination shift” will help clarify and alleviate reader confusion.

To further clarify, we have also added greater detail to the Materials and methods about how this metric is calculated and analyzed – specifically, where individuals are being tracked we include individual ID in our analyses and elsewhere we analyze by group:

“To determine the consistency of gut microbial differences across ordination space due to domestication, *Pan*-*Homo* divergence, or industrialization in the observational study, we calculated the average position of the host dyad (e.g., pig/boar) or all primates (humans and chimpanzees) for axis 1 of the NMDS then measured the displacement along each axis for an individual sample relative to that mean position. We tested for differences in these ordination shifts by domestication status or primate host taxonomy (e.g. chimpanzee versus US human). To estimate the direction and magnitude of changes in beta-diversity during the experimental studies, we tested whether inclusion of a treatment group term improved the performance of a linear mixed effects model relative to a model with only time and animal ID terms for predicting the NMDS1 axis value for an individual. These analyses allowed us to consider the direction of betadiversity changes in addition to the magnitude.”

References for similar ordination analyses: Ang et al., 2020; McKenzie et al., 2017.

3) The authors should be careful with the way they present their results to avoid biased interpretation and make claims that are clearly supported by their results.a) It sometimes seems as if the authors have interpreted the findings to fit a preconceived idea of the findings. For example, the authors conclude a "consistent effect of domestication status" (Results), but the samples cluster by host, which has the highest effect sizes. The conclusion is then mainly based on a statistical analysis that showed domesticated samples to be "further right" on an NMDs axis. This is not very convincing, and not very clear in Figure 1C either.

We have attempted to address the reviewer concerns and remove the impression we are overinterpreting our findings. Specifically, we now start the presentation of our cross-species comparison results by stipulating we don’t find a convergent domesticated microbiota before noting we do see a global signal (not “consistent effects”) of domestication. We have also added PERMANOVA analyses of individual dyads (similar to analyses performed in McKenzie et al., 2017) that substantiate our claim that there are widespread (albeit not universal) effects of domestication.

“Despite observing no single convergent “domesticated microbiota” profile, our analysis detected a global signal of domestication status on gut microbiota composition. Across the combined dataset, the factor that explained the largest proportion of variation was the host dyad (e.g., pig/boar; P<0.001, R^2^=0.39, F=17.086, PERMANOVA; Figure 1B). However, correcting for host dyad, domestication status also contributed significantly to variation in microbial communities (P<0.001, R^2^=0.15, F=6.081), and these results were robust to the distance metric analyzed (Supplementary File 1). Furthermore, analyses of individual dyads found a significant effect of domestication status for all groups except canids (P<0.05, R^2^=0.18-0.41, PERMANOVAs; Supplementary File 1).”

We have moved the dyad-by-dyad breakdown of the ordination shift analysis to the supplement, and now present just the domesticated versus wild ordination shift analysis in the main text and figures to illustrate the overall effect. In addition, we have added discussion of some of the potential reasons for dyads not displaying identical trends when analyzed individually.

“Cases where domestication effects are weaker in our comparative study generally consist of animals where the ecological change associated with domestication has been small—e.g. sheep and pigs, whose diets may be quite similar to their wild progenitors, at least when kept in the non-industrialized agricultural settings that were sampled (9)—or where ecological changes are in the opposite direction from the domesticated norm—e.g. canids, where the domesticate diet typically involves lower protein and higher carbohydrate levels than wild diets, instead of the higher protein levels seen in most laboratory or farm animals (47).”

b) Another claim is that in Figure 2, differences between domesticated and wild mice can be overcome by a diet switch, but looking at Figure 2—figure supplement 2, that is simply not the case. It is difficult to see how the data in Figure 5 provides strong evidence that the effects of domestication and industrialization are similar.

We have addressed this overstatement by rephrasing the Figure 2 legend title to state that “Gut microbial differences between wild and domesticated mice can be partially overcome by diet swap.” We also made similar changes to the Figure 4 legend title, which now reads “Microbial differences between wild and domesticated canids can be partially overcome by diet shifts.” Additional explanation for changes made to address concerns that the effects of domestication and industrialization are overstated can be found in our response to 1a-c.

4) More clarification is needed for wild and domestic microbiome results and subsequent conclusionsa) The results presented (Results and Figure 1) do not seem to support the conclusion that domestication is shifting all species to the right along NMDS1. The magnitude and direction of shift seems to differ based on host species. While the general trend of all species lumped together is to the right, sheep and pigs don't seem to follow the pattern (and some others don't seem to have a strong shift to the right). What are the effect sizes for the Mann-Whitney U tests here? Also, looking at Figure 1—figure supplement 2A, only the companion species are denoted as having a p<0.05, which seems at odds with the statement in the Results.This species-dependent direction and strength of shift is not entirely unexpected based on previous work. Shifting host ecology (diet or captivity) has previously been shown to differentially effect host species: Amato et al., 2015 and McKenzie et al., 2017.The inconsistency in the direction of the shift might not actually negate the broader point, that domestication at times has effects on the gut microbiome that are very similar to the shift we see between industrialized and non-industrialized humans. In fact, it might be instructive to point out what specific species might be good models for the shift we see in humans – what are the specific ecological shifts with domestication in those species and how does that mirror the ecological shift with industrialization in humans?

As noted above, we have made language changes throughout the manuscript to tone down our claims and better clarify our conclusions. In particular, we added separate analyses of wild-domesticated dyads to support the finding of a signal of domestication in the microbiota of diverse lineages of mammals, and have added text discussing potential explanations for variation in the strength of this signal between lineages. For additional detail, please see response to item 3a, above.

The reviewers make a good point that leveraging differences in host species ecology could be a powerful way to understand which ecological aspects domestication and industrialization are most likely to have driven the parallel signatures observed. We will certainly consider such an analysis in the future, but we believe it is beyond the scope of the present manuscript, which serves to launch this line of inquiry by pointing out the existence of a global effect of domestication on the gut microbiota and its parallels with those of industrialization.

b) What does it look like when you put the results of the mouse and canid experiments in the same ordination space with your wild/domestic and chimp/human pairs? Is the shift in the expected direction? When looking at the results of the mouse experiment and the canid experiments on their own, we see a shift to the left with experimental domestication (ie, for the Wildh/Domd treatments), but this might be a function of the ordination space?

While we appreciate the suggestion for combining the comparative and experimental analyses to assess generalizability, we have chosen not to include this analysis in the revised document. A strong signal of species and sequencing run, typical in PERMANOVA analyses and ordination plots of microbiome data, is expected a priori to swamp the experimental domestication signal. The interpretation of left/right shifts is not inherently meaningful as these axes could be mirrored and still present the same data, as such the leftward shift in the experiments does not inherently conflict with the rightward shift observed in the comparative data. We retain these ordination shift analyses because they are inherently more informative than those of dissimilarity (which is directionless) and because the existence of changes along the first axis supports the claim that the effect is meaningful (since the lower the number of the axis, the greater the degree of variation captured). We continue to use the language of left/right in the text to help guide readers in viewing and interpreting the plots, and have provided greater detail concerning the analysis in the Materials and methods to help readers better understand the finding.

c) Were any of the animals, either wild or domestic, from the same family, field, pen, etc.? Cohousing results in convergent microbiome profiles across a number of species due to horizontal microbial exchange. If conspecifics were collected from the same living situation or were related, one might expect higher microbiome sharing on those grounds alone. This potential confounder could explain the high similarity between the conspecifics. These details should be added to the Materials and methods. If this is an issue, it should be corrected for in statistical comparisons (if possible).

Geographic closeness and, especially, cohousing are known to impact the gut microbiota and the reviewers are correct that this may be playing a role here. Unfortunately, due to limits on sample availability, we only have one domestic species where animals were sampled at more than one location. We do have multiple species sampled at the same location although these individuals were never cohoused (or kept in the same pen/field in the case of agricultural animals). To address the concerns of the reviewers, we have added an analysis of location and now include an ordination plot of the comparative data colored by collection locale (Figure S1C).

“We have limited ability to distinguish between locale and species effects since all but one species (sheep) had samples collected from only one locale. Some locales had multiple species present and we do find a significant effect of locale on overall microbial community composition even when correcting for host phylogeny effects (P<0.001, R^2^=0.16, F=6.14, PERMANOVA). However, it is clear that locale does not necessarily lead to convergent microbiota across taxa as evidenced by the low clustering by site in NMDS ordination space (Figure 1-figure supplement 1). When analyzing just the sheep samples, we find a minimal effect of locale (P=0.023, R^2^=0.07, F=2.08, PERMANOVA). ”

5) Technical concerns and data presentationa) Figure 2 and Figure 4 are a difficult to interpret, because the lines used to indicate moving points are obscuring the points themselves in some cases. Would ellipses around the treatment groups in the NMDS plots be more informative than the moving points?

To improve the readability of Figure 2 and Figure 4, we have removed the lines indicating moving points and instead provide the ordination shift plots with individuals plotted separately. Hopefully this allows both to be read and interpreted more easily.

b) For the adonis2 function, to get the marginal sums of squares you need to include “by = "margin"” in the function call. Using adonis2 without specifying “by” is equivalent to using the older adonis function. This should be relatively quick to rerun and will make the effect of host vs. ecology vs. diet easier to parse.

We thank the reviewers for this clarification. We have updated these analyses and the results as reported in the manuscript. There were only minimal impacts of the altered function call, so no interpretations were affected by this update.

c) The OTU picking strategy can introduce biases when sampling microbes that are better represented in the reference taxonomy since more of the sequences will be classified in one sample versus another. Even though the authors seem to have chosen the best option for this dataset, there very well could be differences given that comparisons are explicitly between Industrialized vs Non-Industrialized populations (there tends to be lower read mapping to closed ref OTUs in non-industrial populations) as well as human-associated vs. wild animals (it would be expected that lab animals and livestock microbiomes have been better characterized back when that GreenGenes taxonomy was created).Can a Supplementary file be added that lists the proportion of reads classified per sample? Are there differences in the number of reads that classify between the major comparisons in this paper (Industrialized vs. Non-Industrialized, Wild vs. Domestic, etc.)? If there are, then reprocessing of these reads either with an open OTU calling method or ASV method should be implemented.

This is a valuable point, and we thank the reviewers for raising it. As requested, we have added summary statistics for the proportion of reads classified per sample to Supplementary file 2. We have also tested whether variation in the proportion of reads classified is biased in a manner that might impact our results.

We do observe variation between wild and domesticated samples (P=0.045, Mann-Whitney U test). However, contrary to what reviewers (and we) initially expected, samples from wild animals had, on average, a larger proportion of reads mapped to the GreenGenes database compared with samples from domesticated animals (see Author response image 1 and Author response image 2). There was also variation observed among dyads (P<0.001, Kruskal-Wallis test) but, as far as we can tell, this variation was not patterned based on species ecology.

Regarding the human analysis, we see variation in the proportion of reads classified across populations (P<0.001, Kruskal-Wallis test). However, industrialized and non-industrialized samples had similar proportions of reads mapped, overall. As expected, humans generally have more reads classified than animals, and notably, taxa present in the chimpanzee gut microbiota are substantially less well represented compared with those from all other groups.

**Author response image 2. respfig2:** 

d) How does microbial load/density vary based on gut passage rates, and could this be influencing your results?

We don’t have the data to directly address passage rate here, although relationships between microbial density and transit time have been reported elsewhere in the literature (some citations included below). Broadly speaking, while transit time is likely tied to gut physiology and diet, it is not a static feature of the host. Thus, we did not feel it was appropriate to use published values to infer it and then test for a relationship here. However, to acknowledge this valid point, we now include reference to the possible effect of transit time on gut microbiota composition and load in the Discussion:

“Our reciprocal diet experiments in mice and canids substantiate our claim that ecology plays a predominant role in shaping the domesticated gut microbiota. However, they do not pinpoint the mechanism(s) for these effects. Variability in diet or other aspects of ecology and their concomitant effects on host physiology (e.g., passage rate) can alter microbial composition or abundance through changes in the selective landscape that microbes experience and changes in environmental exposure.”

References for microbial density and transit time: Kashyap et al., 2013 and Vandeputte et al., 2017.